Atmospheric § Chemistry and Physics Discussions

#### Segregation in the Atmospheric Boundary Layer: 1 The Case of OH - Isoprene 2

3 4

Ralph Dlugi<sup>1,2,3</sup>, Martina Berger<sup>1,2,3</sup>, Chinmay Mallik<sup>2</sup>, Anywhere Tsokankunku<sup>2,3</sup>, Michael Zelger<sup>1,2,3</sup>, Otávio C. Acevedo<sup>4</sup>, Efstratios Bourtsoukidis<sup>2</sup>, Andreas 5

6

Hofzumahaus<sup>5</sup>, Jürgen Kesselmeier<sup>3</sup>, Gerhard Kramm<sup>6</sup>, Daniel Marno<sup>2</sup>, Monica 7

Martinez<sup>2</sup>, Anke C. Nölscher<sup>2</sup>, Huug Ouwersloot<sup>2</sup>, Eva Y. Pfannerstill<sup>2</sup>, Franz Rohrer<sup>5</sup>, 8

Sebastian Tauer<sup>2</sup>, Jonathan Williams<sup>2</sup>, Ana-Maria Yáñez-Serrano<sup>3</sup>, Meinrat O. 9

Andreae<sup>3,7</sup>, Hartwig Harder<sup>2</sup> and Matthias Sörgel<sup>2,3</sup> 10

- 11
- 12

<sup>1</sup>Arbeitsgruppe Atmosphärische Prozesse (AGAP), Munich, Germany 13

- <sup>2</sup>Atmospheric Chemistry Department, Max Planck Institute for Chemistry, P.O. Box 14 3060, 55020 Mainz, Germany
- 15

<sup>3</sup>Biogeochemistry Department, Max Planck Institute for Chemistry, Mainz, Germany 16

<sup>4</sup> Universidade Federal Santa Maria, Dept. Fisica, 97119900 Santa Maria, RS, Brazil 17

<sup>5</sup>Institut für Energie- und Klimaforschung: Troposphäre, Forschungszentrum Jülich, 18 Germany 19

<sup>6</sup>Engineering Meteorology Consulting, Fairbanks, AK, USA 20

- <sup>7</sup>Scripps Institution of Oceanography, University of California San Diego, California, 21 USA
- 22 23
- 24 25 Correspondence to: R. Dlugi (rdlugi@gmx.de) and M.Sörgel (m.soergel@mpic.de)
- 26

### 27 28

- Abstract
- 29 30

In the atmospheric boundary layer (ABL), incomplete mixing (i.e., segregation) results in 31 32 reduced chemical reaction rates compared to those expected from mean values and rate constants derived under well mixed conditions. Recently, segregation has been suggested 33 as a potential cause of discrepancies between modelled and measured OH radical 34 35 concentrations, especially under high isoprene conditions. Therefore, the influence of 36 segregation on the reaction of OH radicals with isoprene has been investigated by modelling 37 studies and one ground-based and one aircraft campaign.

38

In this study, we measured isoprene and OH radicals with high time resolution in order to 39 40 directly calculate the influence of segregation in a low-NO<sub>x</sub> and high-isoprene environment in the central Amazon basin. The calculated intensities of segregation  $(I_s)$  at the Amazon Tall 41 Tower Observatory (ATTO) above canopy top are in the range of values determined at a 42 43 temperate deciduous forest (ECHO-campaign) in a high-NOx low-isoprene environment, but stay below 10 %. To establish a more general idea about the causes of segregation and their 44 45 potential limits, further analysis was based on the budget equations of isoprene mixing ratios,

46 the variance of mixing ratios, and the balance of the intensity of segregation itself. 47 Furthermore, it was investigated if a relation of  $I_s$  to the turbulent isoprene surface flux can be established theoretically and empirically, as proposed previously. A direct relation is not 48 49 given and the amount of variance in  $I_s$  explained by the isoprene flux will be higher the less 50 the influence from other processes (e.g., vertical advection) is and will therefore be greater near the surface. Although ground based values of  $I_s$  from ATTO and ECHO are in the same 51 range, we could identify different dominating processes driving  $I_s$ . For ECHO the normalized 52 53 variance of isoprene had the largest contribution, whereas for ATTO the different transport 54 terms expressed as a residual were dominating. To get a more general picture of  $I_s$  and its 55 potential limits in the ABL, we also compared these ground based measurements to ABL 56 modelling studies and results from an aircraft campaign. The ground based measurements 57 show the lowest values of the degree of inhomogenous mixing (

1976; McRae et al., 1982; Donaldson and Hilst, 1972; Donaldson, 1975; Lamb and Shu, 1978). Here,  $c'_i$  and  $c'_j$  denote temporal fluctuations around the mean mixing ratios,  $\overline{c_i}$  and  $\overline{c_j}$ , of compounds i and j, respectively. If for a second-order reaction the product of the mean mixing ratios fulfills the relation  $\overline{c_i} \times \overline{c_j} \gg \overline{c'_i c'_j}$ , the influence of turbulent fluctuating terms in the reaction rate equation  $k_{ij} (\overline{c_i} \times \overline{c_j} + \overline{c'_i c'_j})$  can be neglected for the prediction of either mean value,  $\overline{c_i}$  or  $\overline{c_j}$  (e.g., Danckwerts, 1952; Shu, 1976).

89

If this inequality is not valid, the balances of higher order moments (e.g., variances, 91 covariances, triple correlations) have to be calculated either by the model or by analysis of experimental data. The quotient of the covariance term and the product of the means is 92 commonly called the intensity of segregation,  $I_s = (\overline{c_i' c_i'} / \overline{c_i} \times \overline{c_j})$  (e.g., Danckwerts, 1952; 93 Damköhler, 1957) and is applied to describe the degree of inhomogeneous mixing for 94 95 second order chemical reactions. For this Reynolds-type ensemble averaging of properties of a fluid, the influence of fluctuations on chemical reactions is described by additional 96 97 differential equations to determine the higher-order moments (e.g., Donaldson and Hilst, 98 1972; Donaldson, 1973, 1975; Shu, 1976). Another way to approach this problem is to find the exact properties of the probability density functions of turbulent quantities for each 99 100 reactant (e.g., O'Brien, 1971; Bencula and Seinfeld, 1976; Lamb and Shu, 1978).

The balance equation approach was also applied for the analysis of field measurements of 103 the  $O_3 - NO - NO_2$  system (e.g., Lenschow, 1982; Vilà - Guerau de Arellano et al., 1993; Kramm and Meixner, 2000) and to study segregation of the reaction of isoprene (ISO) with 104 the hydroxyl radical (OH) (Dlugi et al., 2010, 2014). This concept not only considers the 105 determination of first- and second- order moments (mean values, covariances and variances) 106 107 but at least requires the additional knowledge of the third moments – e.g., the skewness, Sk (Stull, 1988; Sorbjan, 1989; Shu, 1976) - to quantify influences of so-called coherent 108 109 structures (e.g., Katul et al, 1997, 2006; Raupach et al., 1996; Wahrhaft, 2000) on segregation (e.g., Dlugi et al., 2014). 110

Modelling studies on segregation for these chemical systems were mainly performed for more complex atmospheric mixtures (e.g., Schumann, 1989; Verver et al., 1997; Vinuesa and Vilà-Guerau de Arellano, 2005; Krol et al, 2000; Ouwersloot et al., 2011; Ebel et al., 2007; Patton et al., 2001; Kim et al., 2016; Li et al., 2016; Gerken et al., 2016) than could be considered by the analysis of field data (Dlugi et al., 2010, 2014; Kaser et al., 2015; Kramm and Meixner, 2000). Recently, the influences of shallow cumulus on transport, mixing and chemical reactions in the ABL were modelled for the reaction *isoprene* + *OH* (e.g., Vilà-

119 Guerau de Arellano et al., 2005; Ouwersloot et al., 2013; Kim et al., 2016; Li et al., 2016).

The dynamics and mixing by shallow cumulus clouds are shown to enhance  $I_s$  also near the

surface at least in qualitative agreement with the scarce experimental findings (Dlugi et al.,

- 2014).

For a characterization of mixing and reaction conditions in atmospheric flows, the Damköhler number,  $Da_c$ , the quotient  $(\tau_t/\tau_c)$  between the characteristic timescales of turbulent or 125 convective mixing processes,  $\tau_t$ , and the specific chemical reaction,  $\tau_c$ , of a compound (e.g., 126 127 ISO or OH), is chosen (e.g., Vilá Guereau de Arellano and Lelieveld, 1998). This 128 dimensionless number allows a classification of  $I_s$  as a function of nearly inert ( $Da_c \ll 1$ ), 129 slow  $(0.05 \le Da_c \le 0.5)$ , fast  $(0.5 \le Da_c \le 5)$ , and very fast  $(Da_c > 5)$  bimolecular reactions with respect to one of the two reactants. Damköhler numbers can be formulated in different 130 131 ways in space and time, depending on the formulation of the turbulence scales (e.g., 132 Donaldson and Hilst, 1972; Molemaker et al., 1998; Koeltzsch, 1998). Therefore, the actual numerical values of Dac in various works found in the literature may differ systematically 133 (e.g., Schumann, 1989; Sykes et al., 1994; Verver et al., 1997, 2000; Li et al., 2016). 134 135 Nevertheless, the ranking of reactions being most influenced by inhomogeneous mixing is 136 consistent within each choice of scales for the calculation of  $Da_c$ .

Some authors applied an additional scaling which uses the turbulent flux of a species 139 (e.g.,  $w'c'_i$ ) at the surface to find a description for the reaction and inhomogeneous mixing (e.g., Schumann, 1989; Verver et al., 2000) and added a second Damköhler number, Daf, to 140 describe the influence of the surface flux on this ranking concept. This approach requires that 141 for a specific reaction (e.g., ISO + OH) the segregation intensity,  $I_s$ , shows a clear functional 142 143 dependence on the corresponding turbulent flux. We will therefore discuss this concept 144 together with the theoretical framework applied to the analysis of the field data in Section 3. 145 In Sections 3.4 and 4.1 we search for theoretical and empirical relations between the 146 turbulent flux of isoprene and the related segregation intensity to test this hypothesis, 147 because some studies suggest that spatially inhomogeneous distributions of emission fluxes significantly influence - in a direct relation - the segregation intensity (e.g., Krol et al., 2000; 148 149 Pugh et al., 2011; Ouwersloot et al., 2011). The results from the aircraft measurements 150 presented by Kaser et al. (2015) are interpreted in this way as well. They even suggest a 151 "feedback loop – the higher the isoprene flux the larger the  $I_s$ ". The analysis of ECHO 2003 by Dlugi et al. (2014) showed that such an influence of spatially variable isoprene fluxes can 152 153 be detected also in the results from measurements near canopy top, but needs a more 154 specific interpretation (section 3.4).

One of the chemical reactions that has been studied experimentally is that of isoprene with 157 OH radicals. Isoprene is an important biogenic compound with a global annual emission of 535 Tg/a to 595 Tg/a (Sindelarova et al., 2014; Guenther et al., 2012). If also the 158 159 dependence on soil moisture stress is considered an annual emission of about 374-160 449 T is estimated (Müller et al., 2008). Isoprene is emitted by various plants (Kesselmeier, 2001; Kesselmeier and Staudt, 1999; Günther et al., 1995). The emission source strength 161 and related fluxes into the atmosphere are mainly controlled by plant physiological factors, 162 absorbed radiation and leaf temperatures (e.g., Kesselmeier, 2001; Guenther et al., 2006; 163 164 Ciccioli et al., 1997; Doughty, Goulden, 2008). After emission, isoprene is mixed by 165 turbulence and convection in a cloud topped ABL (e.g., Heus and Jonker, 2008; Ramos da 166 Silva et al., 2011; Ouwersloot et al., 2013), while being transported with the mean wind field. Isoprene reacts with OH (e.g., Finlayson-Pitts and Pitts Jr., 1986), the so called detergent of 167 168 the atmosphere, which is formed by photochemical reactions and recycled in radical chain reactions (e.g., Finlayson - Pitts and Pitts, 1986; Rohrer et al., 2014). It is a fast reacting 169 compound with  $\tau_c < 1s$ . Therefore, the hydroxyl radical (OH) is only locally determined by 170 chemical sources and sinks which are influenced by the solar actinic flux, ozone  $(O_3)$ , water 171 172 vapor and additional reactants like HO2, NO2, NO, CO, CH4 and various volatile organic 173 compounds (VOCs). We may consider this chemical system in the way that isoprene (with 174  $\tau_c > 300s$ ) is transported through this locally variable field of OH. Furthermore, the variability of the isoprene source strength in time and space (e.g., Ciccioli et al., 1997) - which may be 175 described by the turbulent surface flux of isoprene  $w'c'_i$  - as well as of chemical sources and 176 sinks of OH can contribute to the development of non-homogeneously mixed conditions 177 with  $I_s

193 savanna areas. This inhomogeneity of land surface properties (and of canopy surface 194 temperatures and surface sensible heat fluxes,  $H_s$ ) is related to variations in buoyant production as well as inhomogeneous source distributions for isoprene, both of which have 195 196 an impact on the variability of the isoprene flux and the mixing ratio (e.g., the variance) and, therefore, on  $I_s$  for the isoprene – OH reaction. Comparable to the study by Patton et al. 197 (2001), the modelled chemical reactions are for low  $NO_x$  conditions as found for example in 198 the Amazonian region (e.g., Rohrer et al., 2014) - where one major sink for OH is isoprene 199 200 (e.g., Andreae et al., 2015; Karl et al., 2007; Nölscher et al., 2015; Yáňez- Serrano et al, 201 2015).

202

203 In their LES simulation, Kim et al. (2016) found that  $I_s$  is a function of the  $NO_x$  mixing ratio. They point out that values with  $I_s < -0.1$  both for  $NO_x < 0.2 \, ppb$  and  $NO_x > 1 ppb$  are 204 205 reached in a cloud layer. Positive values of  $I_s$  are calculated in the cloud layer for  $NO_x \approx$ 206 0.5 *ppb*. In the mixed layer of the ABL Kim et al. (2016) found  $I_s < -0.1$  only for  $NO_x \ge 3 ppb$ . 207 Their surface layer (SL) results are nearly independent from the  $NO_x$  mixing ratios with 208  $-0.05 \le I_s \le 0.0$  for a homogeneous isoprene flux. In contrast, near the surface Ouwersloot 209 et al. (2011) found significantly larger values for low  $NO_x$ - conditions, but in a region with 210 inhomogeneous distributions of the surface sensible heat flux  $H_s$  and the isoprene emission flux  $\overline{w'c'_i}$ . In their Fig. 13 they show a case with  $I_s = -0.195$  and  $H_s \approx 0.15 \ Kms^{-1}$  for an 211 212 inhomogeneous distribution of the isoprene emission flux. But most of their results for 213 homogeneous source distributions are in the range  $I_s < -0.1$  and are at least qualitatively comparable to the results of Kim et al. (2016). The experimental values determined above 214 215 canopy top during the ECHO 2003 field study with  $NO_x > 1 ppb$  are in the range  $-0.16 

Dlugi et al. (2010, 2014) analyzed highly time-resolved ( $\leq 0.2 Hz$ ) data from measurements 230 of isoprene and OH during the ECHO 2003 field experiment above a deciduous forest canopy in a polluted area (e.g.,  $NO_x > 1 ppb$ ). They could specify influences of 231 232 inhomogeneous source distribution, turbulence, and cloud-induced convective downward and upward transport on  $I_s$  in the range  $-0.16

ATTO 2015 (see Sections 2.2, 3. and 4.), and NOMADSS (Kaser et al., 2015). In our 264 265 discussion in Section 3 we present the theoretical frameworks, which serve as a rule on how to perform atmospheric measurements of this kind and to analyze the data. As introduction 266 we give the definition of  $I_s$  and explain the different influences of the mean mixing ratios of 267  $ISO(\overline{c_i})$  and  $OH(\overline{c_j})$ , their related standard deviations  $(\sigma_i, \sigma_j)$  and variances  $(\overline{c_i'^2}, \overline{c_j'^2})$ , their 268 covariance  $(\overline{c_i'c_i'})$  and the isoprene flux  $(\overline{w'c_i'})$ . For each of these quantities a prognostic 269 270 balance equation (also named budget equation in the literature) allows us to analyze the 271 impact of different processes on their behavior in time and space (e.g., Stull, 1988; Sorbjan, 1989; Seinfeld and Pandis, 1997). These processes are represented by the different terms of 272 the balance equations as described for the exchange and transport of momentum, heat, and 273 274 moisture for example by Monin and Obukhov (1954), Businger (1973), McBean and Miyake 275 (1972), Panofsky and Dutton (1994), Stull (1988), Sorbjan (1989) or Garrett (1992) and for reacting compounds for example by Shu (1976), McRay et al. (1982), Lenschow (1982) or 276 Ebel et al. (2007). This kind of analysis is done by solving these equations numerically in a 277 model or by calculation of the different terms from direct measurements and order of 278 magnitude estimates based on literature values, as also done in our study. 279

First, we perform such calculations for the balance of the mean mixing ratio  $\overline{c_i}$  based on the 281 data from ECHO 2003 and ATTO 2015 (Section 3.2). Secondly, we discuss the balances of 282 the variances, as they can be directly related to the covariance,  $\overline{c'_i c'_i}$ , and to the segregation 283 intensity, Is (Section 3.3). In the following Section 3.4 we focus on the balance of the 284 isoprene flux,  $\overline{w'c'_i}$ , to analyze if a direct relation to  $I_s$  can be established by a term of this 285 equation, as suggested, for example, by Kaser et al. (2015). Finally, the balance of the 286 287 segregation intensity, Is, itself is evaluated based on measurements. In Section 4 we 288 compare results from earlier modelling studies and direct field measurements near canopy 289 top to each other and to the findings given by Kaser et al. (2015) from experiments in the 290 ABL. The results from experiments in the atmosphere and modelling studies are compared 291 also to obtain some empirical relation between the segregation intensity  $I_s$  and the 292 Damköhler number Dac.

293

280

- 294
- 295
- 296

# 297 2. The Field Studies

### 299 **2.1. ECHO 2003**

300

The ECHO intensive field campaign was performed from 17 June to 6 August 2003 on the grounds of the Reserch Center Jülich, Germany. Three towers were installed in a mixed 302 303 deciduous forest with the dominating tree species, beech, birch, oak and ash, and a mean 304 canopy height  $h_c$  of 30 m. The vertically integrated one-sided leaf area index in a radius of 50 305 m around the main tower varied between LAI = 5.5 and LAI = 5.8. The towers were aligned 306 parallel to the main wind direction (Schaub, 2007) with the main tower in the center. The west tower was located 220 m from the main tower, and the east tower was located 120 m 307 away. This allowed the investigation of the influence of the spatial distribution of biogenic 308 volatile organic compound (BVOC) sources (isoprene, monoterpenes) on measured fluxes 309 310 (e.g., Spirig et al., 2005). The field measurements were supported by the physical modelling of this forest site in a wind tunnel (Aubrun et al., 2005), also to estimate the influences of 311 spatial heterogeneity of emission sources on measured fluxes of some BVOCs. 312

During the ECHO campaign, a feasibility study was performed on 25 July (day 206 of year 315 2003) to measure fluxes and higher order moments (e.g., covariances) not only for isoprene but also for the first time for 0H- and HO2- radicals. The data from these measurements were 316 analyzed in detail for the time period between 09:00 and 15:00 CET. This period was 317 characterized by cloudy conditions with a moderate horizontal wind velocity variation and 318 319 slightly unstable to neutral stratification above the canopy. Broken cloud fields caused significant fluctuations of all radiation quantities above canopy. The air temperature,  $T_a$ , at 320 321 the measuring height  $z_R = 37 m$  increased from 19 to 26.5 °C, while the specific humidity,  $q_a$ , increased only slightly from 09:00 to 12:00 CET from 8.3  $g kg^{-1}$  up to about 9.5  $g kg^{-1}$ 322 and then decreased to about  $8 g kg^{-1}$  (Dlugi et al., 2010, 2014). 323

All measurements reported in the present paper were obtained at the main tower (Dlugi et al, 326 2010; 2014). The main tower with a height of 41 m, and the main measuring platform 327 at  $z_R = 37 m$ , was equipped with nine sonic anemometers/thermometers (METEK, instrument type: USA-1; time resolution 10 Hz) between 2 m and 41 m, and eight psychrometers (dry 328 and wet bulb temperatures) at the same heights, except at 41 m. A time resolution for air 329 temperature,  $T_a$ , and specific humidity,  $q_a$ , of 15 s could be achieved. Radiation quantities 330 and photolysis frequencies were obtained by radiometers directly above the canopy ( $h_c =$ 331 30 m) with a time resolution of 3 s (Bohn et al., 2004; Bohn, 2006; Bohn et al., 2006). 332 333 Occasionally vertical profiles were measured.

The OH and  $HO_2$  radical concentrations were measured by Laser Induced Fluorescence (LIF; 336 Holland et al., 1995, 2003) on a vertically movable platform. For the reported measurements it was positioned above the canopy, with the inlet at 37 m height (Kleffmann et al., 2005). A 337 proton-transfer-reaction mass spectrometer (PTR-MS) for measurements of isoprene, 338 monoterpenes, methyl vinyl ketone (MVK), and methacrolein (MACR) was installed at the 339 ground, using a sampling line to collect air at the height of the ultrasonic anemometer 340 (Ammann et al., 2004; Spirig et al., 2005). The distances of the inlets of the PTR-MS and LIF 341 instruments from the ultrasonic anemometer measuring volume were 0.45 m and 0.6 m, 342 343 respectively. This spatial arrangement requires corrections to the calculated fluxes as 344 outlined by Dlugi et al. (2010) and Dlugi et al. (2014). The time series of 0H (and  $HO_2$ ) and 345 isoprene are available with a resolution of 0.2 Hz for the calculation of higher order mixed 346 moments (e.g., covariances).

349

#### 350 2.2. ATTO 2015

The ATTO-IOP1 was conducted at the Amazon Tall Tower Observatory (Andreae et al., 2015) in November 2015 (from 1 to 23 November) under El Niño conditions (Jiménez-Muñoz et al., 2016; Wang and Hendon, 2017). Measurements were made on an 80 m walk up tower at a height of  $z_R = 41 m$ . The average canopy height,  $h_c$ , in the surroundings of the tower is around 35 m. The vertically integrated one-sided leaf area index (LAI) around the tower was about 6. The land cover in the main wind direction is primary rain forest with an extension of several hundred kilometers. During daytime, cumulus clouds develop regularely after noon.

Isoprene mixing ratios were measured by a PTR-MS at 1 Hz resolution. Air was drawn from the measurement height (41 m) by a 3/8-inch opaque fluorinated ethylene propylene (FEP) tubing at a rate of about 10 l min<sup>-1</sup>. The line was isolated and heated. The inlet was protected by a 5  $\mu$ m pore size Teflon filter. The time delay of the measured signal was corrected by maximizing the covariance between fluctuations of an open path  $H_20$  analyzer (Licor 7500, Licor, USA) in front of the inlet and the signal of the water clusters inside the PTR-MS.

Atmospheric *OH* and  $HO_2$  were measured during 16 – 23 November 2015 using a modified version of the HydrOxyl Radical measurement Unit based on fluorescence spectroscopy (HORUS) instrument (Martinez et al., 2010; Hens et al., 2014). The laser system was mounted on a cantilever balcony assembly at 36 m and the detection systems were mounted

on another cantilever balcony at 40 m along with instruments to measure radiation, isoprene,

- and water vapour. The balcony faced to the north-east, the direction of the prevailing wind.

The measurements of atmospheric OH was achieved by measurements of the total OHsignal, i.e., the signal produced due to fluorescence at 308 nm of atmospheric OH as well as 375 376 of OH produced in the system during its travel time from the inlet nozzle to the detection volume, which we call background OH. The difference between the total signal and the 377 378 background signal is thus a measure of atmospheric OH. The background OH can be 379 measured by scavenging of the atmospheric OH with propane. The propane was introduced 380 through an inlet pre-injector (IPI) mounted on top of the inlet nozzle (Novelli et al., 2014; Mao 381 et al., 2012; Hens et al., 2014). During previous campaigns using the IPI system, the propane flow was switched on and off for two minutes each, providing a 4 minute time 382 383 resolution for measurements of atmpospheric OH. For this campaign, we used an additional 384 detection unit for simultaneous measurements of total and background OH in order to 385 increase the time resolution of atmospheric OH measuremets. The detection unit for 386 background OH was placed 55 cm to the east of the detection unit for total OH (Figure 1).

387