# Peer review of "Atmospheric § Chemistry and Physics Discussions"

_Atmospheric Chemistry and Physics, 2018_

## Referee Comment (RC1) · Anonymous Referee #1 · 3 Jun 2019

Dlugi et al. present new data on the issue of segregation between isoprene and OH, and put these in context of previously published work. Their data were obtained in the Amazon and largely confirm previous studies on segregation. The paper could provide new insights on the topic of segregation in the Amazon, but there are a couple of issues that should be addressed before any possible publication.

The manuscript is unnecessarily long (60 pages) - as it stands, the manuscript could be significantly shortened and more focused on the important findings. Vast parts read like a review article. Detailed descriptions of previous studies (section 2.1.) could be significantly reduced and rather be included as a citation. The lengthy discussion of previous studies keeps the authors from describing important details about the ATTO site itself, which is essential to the interpretation of the presented analysis. Section 2.2

therefore lacks clarity.

What is the immediate footprint of the surroundings? What is the main wind-direction, is there a variation in isoprene emissions surrounding the site? Have the authors looked at sector dependent isoprene emissions? From their assumptions it appears the site is characterized by a rather homogenous isoprene emission source, but it would be good to show this. What QAQC criteria were incorporated for the interpretation of turbulence measurements? I would have also expected to see an overview on latent, sensible and momentum fluxes as well as other important micrometeorological quantities such as Bowen ratio, Obukhov length etc.

The key instrumentation relevant to this article are HOx and isoprene measurements. The frequency of isoprene measurements was 1 Hz, so one would expect a loss in high frequency variability. Further, damping through a 40+ m line has to be expected. The method of inferring a lag-time by comparing water vapor fluctuations through such a long line bears a potential problem, because water vapor retention is expected to be much larger than that of small hydrocarbons. It is also not clear what pressure drop was produced by the 5 um filter. One way to ensure that this analysis is not prone to any substantial bias would be to compare the covariance functions between vertical wind, isoprene and water. I am also missing information on the determined delay/lag time. Overall I am concerned that some (significant?) part of isoprene variability might have been lost due to the experimental setup? Have the authors done any co-spectral analysis?

The authors present the issue of underestimating modelled OH in the tropical atmosphere as a main cause to look into the subject segregation. There are some reports of a possible overestimation of OH inferred from LIF instruments. Several recent studies (e.g. Liu et al., 10.1126/sciadv.aar2547 2018) have concluded that there is no gap between modelled and observed OH in Amazonia within the experimental uncertainty. The cited study by Kaser et al. actually also shows this, as the total impact of different chemical recycling schemes in their study seemed to be quite small. It would

strengthen the manuscript to point out differences in OH measurements during this and previous campaigns, as well as commenting on conclusions of the above papers. In this context it is not clear whether there have been any changes to the presented LIF OH measurements since Lelieveld et al., 2008. At least a reference to a recent validation or intercomparison paper would be warranted. A recent chamber study (Kanaya et al., 10.5194/acp-12-2567-2012, 2012) suggests about a 50% uncertainty (bias) for the measurement of OH in low NOx, isoprene dominated environments. If for example LIF OH measurements were subjected to an offset problem, it would probably not impact the presented analysis of this paper, but if there was a problem associated with a sensitivity bias it certainly would. I am wondering whether this could explain some of the different trends shown in Fig. 4.

The derivation of some of the simplifications is poorly explained – eq. 19: why would only one triple term be important in the analysis here? RES, RE and REis are not well explained – I assume REis refers to term I in eq. 21. In general, I miss a thorough analysis of error propagation in context of the presented equations (e.g. eq 21). Many terms are dropped because they are supposedly small, yet the impact of the experimental limitations is not rationalized well in context of the variance budget of isoprene. I suspect that a significant amount of variance of isoprene might not be accounted for due to spectral attenuation. It also appears that the data availability is rather thin – I only count about 16 individual data points for the analysis presented in Fig3. Within the uncertainty of the analysis, I wonder whether this is enough to draw some of the presented conclusions after considering a thorough analysis of the propagation of errors (ie. systematic and random).

---

## Referee Comment (RC2) · Ian Faloona (Referee) · 6 Aug 2019

Review of "Segregation in the Atmospheric Boundary Layer:  The Case of OH - Isoprene" by Dlugi et al., submitted to ACPD December 2018.

Overall Recommendation:

This manuscript attempts to bring together all available measurements (and one modeling study) of the intensity of segregation between OH and isoprene in the convective atmospheric boundary layer (CBL). The objective is not only to better understand this parameter but to present some possibilities for universal parameterization of this coefficient, which directly influences chemical reaction rates based on mean concentrations as applied in most atmospheric chemistry models. The manuscript provides the theoretical development and scale analysis of the most pertinent turbulent statistics (mean concentration, vertical turbulent flux, scalar variance, and reactive chemical trace gas covariance) that underly the physics and chemistry of the segregation coefficient, $I_s$.  This is a very helpful exercise, although half of these equations have been covered in the previous work of Dlugi et al. (2014) – namely, the mean concentration and covariance budget equations.  Nevertheless, to have it all done in one place and to include scaling values from the two tower data sets, ECHO & ATTO, is in my opinion the most valuable part of this manuscript as it exists now (although this has already been done for ECHO in Dlugi et al. (2014). The inclusion of the modeling work and the two airborne flight legs are not similar enough to warrant inclusion in this work because the conditions are different and the behavior of $I_s$ is not understood well enough to simply extrapolate them along any of the proposed independent variables presented in this paper. While I believe there is a lot of this work that is worth publishing I believe that the effort to cull it and improve it to the point of acceptance might take more time than the typical review turnaround, and for that reason I recommend to not publish this work in its present form.

The two central faults I find with the work as presently constituted are:  1) there is a troubling inconsistency with negative signs in this work that appears in Dlugi et al. (2014) also, which absolutely needs to be corrected before even considering publication, and 2) the variability in $NO_x$ concentrations of the different environments are mostly ignored in this work and, in my opinion, are likely to dominate the variability of the data sets reviewed here. On the latter point, I realize that there is *some* justification for ignoring the $NO_x$-dependence based on Kim et al. (2016); however, their simulated segregation coefficients for the surface layer are only about -0.02, which does not agree very well with the near canopy data sets presented here, and their modeling work also shows significant $NO_x$-dependence in other parts of their domain ("mixed" and "cloud" layers, and even changing sign.)

 Some other, less important but serious weaknesses of the submission include:

1)  There needs to be consistency throughout the manuscript in the use of "larger" and "increase" when it comes to the covariance which, in principal, can be either positive and negative. Sometimes it seems to refer to greater magnitude (larger negative number) and sometimes greater numerically, and it often confuses the discussion. [see individual comments for specific points in the text.]

2) This manuscript is way too long, and its excessive length is not justified. I suggest removing the inclusion and discussion of Kaser et al. (2015) which is in the bulk of the CBL and as such is not easily compared to the other two experiments just above forest canopies. I also question the similarity of the results of Ouwersloot et al. (2011) that are averaged over the depth of the CBL when the data from ECHO & ATTO are in the roughness sublayer of large forest canopies.

3) Much of this work is a direct reworking of the results presented in Dlugi et al. (2014) including many of the figures (the latter's figures 21, 20, 16, 9, 8). I realize the utility in updating these relationships with the most extant data possible, but there is no indication that all of these figures contain much information. For example, I do not see any convincing relationship between $I_s$(OH, Iso) and the surface buoyancy production rate, or the correlation coefficient, $r$(OH, Iso), or the Damköhler number. Thus the authors might want to consider which relationships show the most informative new information and only update those.

Specific line-by-line concerns:

Lines 40-43: Please give an approximate range of $NO_x$ for these conditions (instead of "high $NO_x$" and "low $NO_x$" which probably mean different things to different readers.) As stated above, I believe that it may be the most important distinction among the various experimental results.

l. 48-49: I disagree, a direct relation is shown in your equations (11) and (21). The isoprene flux is contained in two leading terms: one is in the TPI of the covariance budget and the other is in the variance budget ($GP_{var}$). But, as you say here, they may be more or less influential depending on the strength of other competing terms.

l. 51: I realize that this may seem like quibbling, but chemical scalar fluxes do not necessarily always decrease with height. For example, if the entrainment is strong and the CBL concentration high enough, then it is possible for the isoprene vertical flux to *increase* with height (e.g. water vapor fluxes in some cases.)

l. 58: You should probably be more quantitative about this statement. How much does $I_s$ increase with measurement bandwidth? It seems that if this is one of the leading findings of the study, worthy of inclusion in the abstract, then it should be elaborated a bit more: what does the cospectrum of  look like at low wavenumbers? The increase in lower frequencies that you investigate in this study goes from about 2-10 km (based on a wind speed of about 4 m/s), comparable to the LES domain of Ouwersloot et al. (2011). On the other hand, the scales covered in Kaser et al. (2015) start at about 3 km (30 s OH measurements) and run to 50-100 km, which are dramatically larger scales than even your expanded analysis. This is the main reason that I believe the inclusion of the results of Kaser et al. (2015) is not appropriate for this work.

l. 70: "and" is probably more often the case (buoyant & shear production combined).

l. 160: missing 'g' in Tg.

l. 165: I don't see why the transport has to take place in a cloud-topped boundary layer.

l. 172: I don't think that $NO_2$ generally determines the OH reactivity in any significant way. Also, whenever there is any significant isoprene, it tends to be the dominant VOC sink for OH. Therefore it doesn't make sense to consider isoprene moving through a static OH field because isoprene *is* usually the dominant sink and determines, in part, the OH field. Of course, this rapid reactivity is what drives the anticorrelation.

l. 178: Doesn't $I_s < 0$ hold only for when the chemical sink of OH is dominated by isoprene or something correlated with isoprene? If isoprene is correlated with a major source of OH (e.g. $RO_2$ or HCHO) then $I_s$ could be $> 0$ in principle, no? Come to think of it, this is where I believe the answer to the origin of your observed $|I_s|$ limitation lies: the OH photochemistry is so heavily buffered that while isoprene is a dominant sink it is also correlated with important sources such as isoprene peroxy radicals, Iso-O2, and HCHO. See Kaser et al., 2015 Fig. S9 for estimates of the relative magnitudes of $RO_x$ (=$RO_2$ + $HO_2$) source strengths.

l. 188: 'caused by' is an odd way to put it because $I_s < 0$ is by definition represents an anti-correlation, but what actually caused it is the broader question that this paper tries to address.

l. 207: The $I_s$ values of Kim et al. (2016), albeit very small, nearly double across the range of $NO_x$ from the experiments you compile (~0.1 from ATTO to ~2 ppb from ECHO) from about -0.02 to -0.035.

l. 209: The UNO run of Ouwersloot et al. (2011) developed an $I_s$ of -0.12 and it was 'homogenous' in heat and isoprene fluxes, whereas without $NO_2$ (lower $NO_x$) the control run $I_s$ = -0.07. Do you mean $I_s$ (low $NO_x$) $> I_s$ (high $NO_x$) or their magnitudes? Note that Ouwersloot et al. (2011) (from their Section 3.6) "stress the need to take the VOC **and $NO_x$** conditions into account in future studies that aim at segregation parameterizations." This advice seems to have been overlooked in the present work.

l. 213: Again, I disagree with the statement that most of the results of Ouwersloot et al. (2011) are $< -0.1$. From their Table 4, HOM $$ = -0.07 which is definitely *not* $< -0.1$! (see less important weakness (1) above about comparisons of $I_s$ magnitude or numerical values less than zero.)

l. 243: I think you should define this here: $<w'c'^2>$. At this point I did not have any idea what $M_{21}$ represented. Incidentally, this appears to be the best predictor you have observed to correlate with $I_s$, so why not emphasize that more and show the results for $M_{21}$ vs. $I_s$ in the ECHO & ATTO data sets?

l. 255-257:  I think you should point out here that most theoretical treatments show $I_s$ to be of smaller magnitudes in the bulk of the CBL vs. the surface layer (e.g., Kaser et al., 2015; Ouwersloot et al., 2011; Patton et al., 2001.)

l. 316:  The measurement of $HO_x$ fluxes and other higher moments from July 25 of the ECHO campaign seems like new material (not covered in Dlugi et al., 2014) so it probably merits some more explication.  For instance, how did the cospectra compare to <w'T'> or <w'O$_3$'> or some other scalars? Does the flux direction and magnitude agree to theoretical predictions (e.g. Gao & Wesely, 1994)? Were these data analyzed in a separate paper?  What changed to allow for these measurements, of which I know of no others?  See  Section 5.1 of Dlugi et al. (2014): "Spatial derivations of mixing ratios of these compounds and their fluxes are not available from this data set". What has changed?

l. 321-323:  What seems more directly important than temperature and humidity is the mean concentrations of $HO_x$ on that day relative to the rest of the experiment.

l. 345:  If the $HO_x$ data is available at 0.2 Hz, why were there no fluxes reported other than July 25?

l. 378:  What was an average background OH value relative to the total?  And as far as the second unit is concerned, how are you sure the amount of background OH is the same in both units? It seems that it might be worthwhile to describe some statistics of the backgrounds for both units to understand their variability and similarity.

Figure 1:  This figure is probably not necessary since none of the analysis of the OH measurements is covered herein.

l. 453:  Equation 4 is a few pages later than this reference, but come to think of it equation 4 does not really give any chemical information other than there is a reaction between OH and isoprene, and as such is probably not necessary.

l. 457:  The justification of time resolution invoked by Karl et al. (2013) is for turbulent fluxes (from Lenschow & Kristensen, 1985), which relies on assumptions about the turbulent statistics of $w'$, but the requirements for a covariance with another scalar are different.

l. 489-491:  I don't think it makes sense to mention compressible fluids and refer to the various '$\alpha$' variables defined by Richardson only to redefine them.

l. 500:  I do not think that '$\times$' is the best symbol to use for multiplication?  It looks like a cross product and the subscripting $i,j,k$ looks like tensor notation. I think a '$\cdot$' or nothing at all is much more conventional and clear to indicate scalar multiplication.

l. 507:  Very rapidly after the H abstraction of the OH + isoprene reaction the production of some isomer of an Iso-$O_2$ radical occurs. These peroxy radicals reacting with NO are usually a

very important source of OH (Kaser et al., 2015 estimate it to be of similar magnitude as ozone photolysis.)

l. 508: Is this supposed to mean Eq. 2 subtracted from Eq. 1?

l. 515: What exactly is $C_xH_yO_z$? I recognize the attempt to remain general, but I think the more important general species that is not mentioned anywhere is $RO_2$.

l. 529: the denominator of term 1 should be $r_{wci}$.

l. 574: Using the Einstein summation convention with tensor notation confuses your own convention of using a generalized chemical trace gas, $c_i$ and $c_j$. These look like 3D vectors in tensor notation ($i, j$ = 1, 2, 3), and there is no other place in the manuscript where it is beneficial to generalize the chemistry. This is a work centered on the reaction of OH and isoprene and as such there is nothing gained by calling those species $c_j$ and $c_i$, respectively. For example, $k_{ij}$, looks like a second order tensor, not a scalar reaction rate coefficient.

Table 1: It would be more clear if you preserved the signs of these terms such that MR & TR are always <0 (that is, act to reduce $<c_i>$ in the budget.) Realizing that you labeled the terms inside the parentheses in equation (7), the terms DMF and DTF have different units because they are not the divergence thus the numbers in the table do not have to sum. I think it would be a lot clearer if you defined the terms to each entire item in the budget equation and keep their signs clear and indicative of how they change $<c_i>$.

l. 603-628: This entire paragraph suspiciously omits any mention of horizontal components of these advective terms. That, in itself, seems like an oversight, and it renders the last sentence (l. 626-628) incorrect: a change in mean horizontal advection (without a change in the wind field divergence) can lead to significant changes in $<c_i>$. It is not clear whether you mean 3D or 2D divergence/convergence in this discussion. Keep in mind that the second term in equation (8) in its full 3 dimensions is zero because of the incompressibility of the mean flow.

l. 707: Doesn't the concentration of isoprene *decrease* with height directly above the canopy making the numbers you report -0.01 to -0.07 ppb m$^{-1}$? This term then is always negative acting to reduce the steady-state variance, no? This is counterintuitive, but is the nature of your steady-state approximation in equation (11).

l. 717: I think you should be more specific in this reference. I believe it is specifically discussed in Section S3.2 of the Dlugi et al. 2014 supplementary materials.

l. 733: Fig. 1 of Spirig et al. (2005) indicates a tower separation of ~250 m.

l. 740-746: $TT_{var}$ is the *divergence* of a turbulent flux of variance. Speaking of a "vertical change of $TT_{var}$" sounds like you are now looking at the second derivative of the variance flux. Is that

what you're referring to?  It would help if this discussion were a lot more clear about what is a turbulent flux of variance (the $<w'c'^2>$ term), vs. its vertical change ($TT_{z,var}$).

l. 752:  Your term III in Eq (11) is equivalent to IV3 in Table 4 of Dlugi et al. (2014) which states its estimated magnitude as < 3e-5 ppb$^2$ s$^{-1}$.

l. 762:  It is not clear how you estimate the gradient of a fluctuating scalar directly, but in general variance budget discussions the molecular destruction term is expected to be first order (to balance mean gradient production in the steady-state, flow-integrated condition.) See Section 5.3 of Wyngaard (2010), for example.

l. 805:  How did you derive these OH flux values?  And are you imply that you have these values for both experiments? Is this not discussed anywhere else in the literature? It seems like a very difficult measurement to directly make by eddy covariance. In any event, you should probably specify the sign of this flux (I believe it should be downward, <0). These magnitudes seem much larger than predicted by Gao & Wesely (1994).

l. 834/5:  Again, the gradients of isoprene should be negative.

l. 845:  I recommend sticking to a single format for all of these range limits of your scale analysis, and preferably using only one significant digit.  For example, '$x$e-3' to '$y$e-1'. Two significant digits for these scale analyses that typically span multiple decades just seems unnecessary and slightly confusing.

l. 859-861:  The similarity you are applying to associate the different scalar transport terms needs to be explicitly stated.  It seems like you are using some sort of modified Bowen ratio analog to the transport term, but this seems highly speculative.  I believe that speculative is fine, but it would be more convincing if you explicitly stated the similarity you are applying.

l. 893:  This range of a factor of 5 for the pressure transport term implies that the time scale values have a range of a factor of 6, because the isoprene fluxes mentioned above span a factor of 30 (0.02 to 0.6 ppb m/s).  It would be clearer if you presented what the mixing length concept of Poggi et al. (2004) depended on.

l. 973:  I have tried and tried and redone the arithmetic on the governing equation (15), because I know how pernicious and elusive sign errors can be, but I just cannot see how the normalized variance term in equations (20 & 21) can have the opposite sign of the $C_{ij}$ term (which is the balance of the terms from $R_{ij}$ outside of the covariance and variance terms that all have the same sign). This same error appears in Dlugi et al. (2014) at their equation (15). This has very important bearing on the analysis because the normalized variance of isoprene **and** the RES (Eq. 16) terms both act to **increase** the magnitude of the OH and isoprene segregation coefficient, in this case, - $I_s$, because $I_s$ < 0.  It seems like this equation will change the authors' calculations of $RE_{is}$ because they solve for it as the residual of equation (21), and will fundamentally change Figure 7.

l. 999:  You say that $R_{ij}$ goes *negative* despite terms (b) and (c) which are positive definite.  But $R_{ij}$ is defined with a negative (definite) sign (equation 18), so the positive definite terms like (b) and (c) work to make $R_{ij}$ negative.  I found this language error typical throughout the manuscript.  When revising I recommend being very careful with the language about discussing relative values or magnitudes of values, always retaining the accurate signs of the terms.

Figure 3:  60% of graph has no information on it. Also, why is the total term in one unit (ppb$^2$ s$^{-1}$) and the individual components in another (ppb$^3$)?  I think it makes the figure less clear to include the reaction rate in one and eliminate it in the others.

Figure 4:  Again, why compare these terms of differing units and then put a one-to-one line on the figure?  Also why ignore the sign of $R_{ij}$? If all the values are negative, then label it -$R_{ij}$.

l. 1041:  Term (c) is not the only leading term of $R_{ij}$.  The ATTO results could differ because of a substantially different contribution from the <OH'Iso'>[Iso] term (a), especially at higher values of [OH]var(Iso).

l. 1109:  "$R_{ij}$ increases [**in magnitude**] with increasing variance…"

l. 1126:  It does not seem clear from Fig. 9 that the relationship between $r_{ij}$ and $I_s$ is *non-linear*. You use this term a lot but none of the figures clearly show any distinction among a linear or non-linear relationship.

l. 1219-1221:  You do not know for certain that the var(Iso) and flux terms are *only* established near the surface (for example, the entrainment zone can possess high variances and fluxes.) Furthermore, equation (11) also shows that var(Iso) is augmented by a term proportional to [Isoprene]<$c_i \cdot c_j$> (concordant with equation (21) with the corrected sign), which could also be a leading term near the surface. Also note that what you are referring to as $GP_{var}$ actually serves to decrease isoprene variance in the steady-state form you present in equation (11) because $d[Iso]/dz$ < 0. In the variance budget, equation (9), $GP_{var}$ produces variance, but in the reactive chemical steady-state of equation (11) it reduces variance.

l. 1243:  I am assuming you mean vertical advection by the mean flow. However, just because $W$ is larger in magnitude at higher elevation in the CBL does not mean that the magnitude of the scalar gradient is larger. It is much more likely to be turbulent transport that is a large term. If by 'vertical advection' you mean turbulent transport (the divergence of a vertical turbulent flux), then I would specify that.

l. 1244:  $I_s$ is related to the isoprene flux by two separate terms of Eq. (21): the *TPI* term of $RE_{is}$ *and* the $GP_{var}$ term in the normalized variance, nvar(Iso).  This is not made clear in this discussion and consequently these arguments are ambiguous. These two flux terms have different coefficients (OH and isoprene gradients, respectively) so that their coefficients will change with altitude (probably both decreasing with height.)  I would suggest eliminating all of

this height dependence of variance discussion because it is speculative (for reactive scalars) and it does not really help the overall work in any way that I can discern.

l. 1319:  No, OH and $O_3$ do not *necessarily* have a large positive covariance (presumably someone could check if there were $O_3$ fluxes being measured on the tower), but the principal source of OH (on the ~1 s time scale) is the photodissociation of $O_3$ so it is very likely that they are, in fact, correlated.

l. 1331-1333:  That is patently incorrect. First, $E_{i0}$ **is** directly related to the flux at any height in the CBL (you used such a relationship yourself earlier to extrapolate their observed fluxes at $z/z_i$ ~ 0.4 to the surface). Furthermore, as stated previously, $I_s$ is correlated to the isoprene flux through both the $GP_{var}$ (where it serves to diminish the variance, and thus $|I_s|$, right above the canopy where the flux is upward and the gradient is negative), and in the TPI term of $RE_{is}$ in (21) where it tends to be a source of negative covariance because the OH gradient is likely positive (due to preponderance of sinks effusing out of the canopy.)

Figure 12:  Why are there are not the same number of blue diamonds (spectrally extended) as there are black circles? They should be 18-27% larger according to line 1391. Also, the blue diamonds all lie exactly on top of the circles showing no spectral change in <OH'Iso'>. Also, the two points on the lower left ($I_s$ < -0.2) do not seem to exist on Fig. 14.

Can you explain what the blue dashed curves represent?  Are they a power fit with n=2 and n=3?

60% of this central figure has no information on it aside from a legend.

l. 1409-1410:  $I_s$ never becomes independent of [Iso][OH] because that product resides in its denominator. The covariance may become independent, but not $I_s$.

l. 1460:  According to Dlugi et al. (2014) Eq. (17) $M_{12}$ were considered the "ejections", and $M_{21}$ the "sweeps"?

Figure 14:  This figure is nearly identically the same as Dlugi et al. (2014) Figure 20, save for the three modeling results and two Kaser et al. (2015) points.  Why do you not present any of the ATTO data on this figure?  Why plot both BP and kinematic heat flux?  As far as I can discern there is no appreciable difference in the underlying relationship and plotting both just clutters the figure.

l. 1545:  It is very challenging to find an empirical relationship in Fig. 14 as stated. You should propose one if you think it exists.  Is looks to me like a nearly vertical line would fit through the points of BP > 3e-3?  I wonder what the *p*-value of such a fit would be, because it does not look great by my eye.

l. 1597-1601:  If $M_{21}$ vs. nvar(Iso) & $RE_{is}$ shows a strong relationship as in Fig. 18 of Dlugi et al. (2014) why not show it?  If this finding is worthy of a paragraph in conclusion, then it seems it should be represented in a figure. Earlier you state the sweeps only weakly correlate with nvar(Iso) and $RE_{is}$, and here you state that only ejections contribute to $I_s$.  This all seems to beg for a figure of both $M_{21}$ and $M_{12}$ to see how much they each correlate to nvar(Iso) & $RE_{is}$. This could be a micrometeorological parameter that is readily measured in canopy field studies that could be used to estimate $I_s$ for chemical modelers, for example.

l. 1621:  The bandwidth of the Kaser et al. (2015) measurements were out to nearly 100 km. For typical winds speeds of, say, 5 m/s this would require a 5.5 hr integration time at a tower site. Thus the measurements, aside from being made several hundred meters higher than the ECHO & ATTO datasets, represent a much larger spectral band.

The 'hypothesis' of scale dependence is established explicitly in Ouwersloot et al. (2011), why bring this in as a hypothesis from this work? There is currently no easy way to disentangle the isoprene surface source variability from the scale of the measurements in terms of their effects on $I_s$, so it is not a hypothesis that is truly tested in this work.

l. 1625-1627:  This is an interesting idea, but not very well explicated in the body text of the manuscript, and only sprung on the reader in the last sentence of the work. The diurnal source correlations (which in and of itself would promote a *positive* species covariance) occur on long time scales relative to the chemistry and the TKE dissipation and the 10-40 minutes averaging used in this study.  In order for this to be a reason for the "limits" of $I_s$ suggested on the 10 min scale the sources would need to correlate on this restricted time scale, and/or there would need to be some sort of downscale cascade at play. This speaks to the absence of any cospectral representation of $I_s$ in this work (something like Fig. S4 of Kaser et al., 2015), which would help understand its spectral dependence. In any event, I suspect the compensating chemistry of OH sources that are correlated with isoprene (e.g. isoprene peroxy radicals) are the most likely culprits for limiting the magnitude of $I_s$.

Reference Su et al. (2015) I was not able to find.  It seems better to use:  Atmos. Chem. Phys., 16, 7725–7741, 2016. www.atmos-chem-phys.net/16/7725/2016/. doi:10.5194/acp-16-7725-2016.

---

## Author Comment (AC1) · 20 May 2020

We thank the reviewer for his helpful comments helping to improve the manuscript.

*Rev: Dlugi et al. present new data on the issue of segregation between isoprene and OH, and put these in context of previously published work. Their data were obtained in the Amazon and largely confirm previous studies on segregation. The paper could provide new insights on the topic of segregation in the Amazon, but there are a couple of issues that should be addressed before any possible publication. The manuscript is unnecessarily long (60 pages) - as it stands, the manuscript could be significantly shortened and more focused on the important findings. Vast parts read like a review article. Detailed descriptions of previous studies (section 2.1.) could be significantly reduced and rather be included as a citation. The lengthy discussion of previous studies keeps the authors from describing important details about the ATTO site itself, which is essential to the interpretation of the presented analysis. Section 2.2 therefore lacks clarity.*

In general: We propose to add a Supplement to the revised version of the paper. Here we give figures S1 – S27 for this Supplement as an example to provide information on the micrometeorological and chemical situation at the ATTO site on day 22/11/2015.

Ans: Indeed, it is a review of available results on measurements on segregation for the OH- isoprene system with comparison to available information from modelling. This is described in section 1 and specified in lines 262 – 292.
The results obtained during the ATTO 2015 study are compared to available field studies (ECHO 2003, NOMADSS 2013) and results from modelling. In addition, the revised version will be extended to discuss the possible impact of NOx on segregation intensity in the field and for model results. See also general comments to reviewer 2.
Therefore a short description is given for ECHO in section 2.1 with references to Dlugi et al. (2010) and Dlugi et al. (2014).
We agree that the description of ATTO 2015 is too short with respect to the remarks given in this review. Therefore we add a) additional information in section 2.2., and b) in a Supplement (as done by Dlugi et al. 2014) for general features on micrometeorological and chemical quantities. Graphical examples for the description of micrometeorological and some chemical quantities are given in Figures (Supplement) S1 – S27.

*Rev: What is the immediate footprint of the surroundings? What is the main wind-direction, is there a variation in isoprene emissions surrounding the site? Have the authors looked at sector dependent isoprene emissions? From their assumptions it appears the site is characterized by a rather homogenous isoprene emission source, but it would be good to show this. What QAQC criteria were incorporated for the interpretation of turbulence measurements? I would have also expected to see an overview on latent, sensible and momentum fluxes as well as other important micrometeorological quantities such as Bowen ratio, Obukhov length etc.*

Ans: Note that we did not make assumptions related to the source distribution of isoprene or other quantities. Instead, we pointed out in section 3 that for ECHO 2003 and ATTO 2015 the influence of advection on the flux as well as also on the variance of isoprene is not negligible (see for example Table 1 and Table 2). Footprints are given in Fig. S6 – Fig. S7 and Fig. S24 – Fig. S25 (see Supplement).
The QAQC criteria that were applied to the field data from ATTO 2015 and ECHO 2003 will be described also in the Supplement, although they are given in sections 3.4 and 3.5 of Dlugi at al. (2010) and by Pfannerstill et al. (2018) for ATTO 2015.
In general all studies in the field are made "at a tower in forests", and therefore are "point measurements", necessitated by the fact that the related instrumentation is very complex and can be operated in a reliable controlled manner only in a way to obtain time series from one measuring volume. Spatially resolved – simultaneous measurements of this kind

at different locations – were not performed up to now. The airborne measurements reported by Kaser et al. (2015) were along a transect sample from a certain volume above ground. They obtain a spatial (and time) average. As for all micrometeorological data processing procedures, the assumption is made that spatial and time averages agree. (See lines 482 - 496 in our paper.)

*Rev: The key instrumentation relevant to this article are HOx and isoprene measurements. The frequency of isoprene measurements was 1 Hz, so one would expect a loss in high frequency variability. Further, damping through a 40+ m line has to be expected. The method of inferring a lag-time by comparing water vapor fluctuations through such a long line bears a potential problem, because water vapor retention is expected to be much larger than that of small hydrocarbons. It is also not clear what pressure drop was produced by the 5 um filter. One way to ensure that this analysis is not prone to any substantial bias would be to compare the covariance functions between vertical wind, isoprene and water. I am also missing information on the determined delay/lag time. Overall I am concerned that some (significant?) part of isoprene variability might have been lost due to the experimental setup? Have the authors done any co-spectral analysis?*

Ans: The instrumentation to measure isoprene during ATTO 2015 is also described by Pfannerstill et al. (2018) as cited above. We performed the requested calculations of correlations before the field study in many laboratory tests as well as on data obtained 2015 in the field, e.g., on water vapor, $CO_2$, and isoprene. The concern "that some part of isoprene variability might have been lost" is surely correct for frequencies above 1Hz. Note that we focus on segregation between isoprene and OH. For OH we apply the time resolution of 15 s, so that all covariances $\overline{OH'ISO'}$ are for $0.06\overline{6}$ Hz (0.067 Hz) and smaller for ATTO 2015 and for ECHO 2003 are for 0.2 Hz and below as given on Page 42, lines 1308 – 1313 in our paper. For these frequency ranges the calculation "of ogives for $\overline{OH'ISO'}$ shows that the loss of variability" is not as important as the loss on the low frequency end of the time window being chosen, as was also discussed by Dlugi et al. (2014) (their Fig. 17). Note that averaging over updrafts and downdrafts may result in small Is as the contributions partly compensate each other (see also their example given in their Fig. 17)
In a Supplement we will give more details on these aspects for ATTO 2015 and also summarize such results for ECHO 2003.

*Rev: The authors present the issue of underestimating modelled OH in the tropical atmosphere as a main cause to look into the subject segregation. There are some reports of a possible overestimation of OH inferred from LIF instruments. Several recent studies (e.g. Liu et al., 10.1126/sciadv.aar2547 2018) have concluded that there is no gap between modelled and observed OH in Amazonia within the experimental uncertainty. The cited study by Kaser et al. actually also shows this, as the total impact of different chemical recycling schemes in their study seemed to be quite small. It would strengthen the manuscript to point out differences in OH measurements during this and previous campaigns, as well as commenting on conclusions of the above papers. In this context it is not clear whether there have been any changes to the presented LIF OH measurements since Lelieveld et al., 2008. At least a reference to a recent validation or intercomparison paper would be warranted. A recent chamber study (Kanaya et al., 10.5194/acp-12-2567-2012, 2012) suggests about a 50% uncertainty (bias) for the measurement of OH in low NOx, isoprene dominated environments. If for example LIF OH measurements were subjected to an offset problem, it would probably not impact the presented analysis of this paper, but if there was a problem associated with a sensitivity bias it certainly would. I am wondering whether this could explain some of the different trends shown in Fig. 4.*

Ans: In our studies and the related paper we are analyzing data on the mean reaction rate between OH and isoprene $kij \cdot OH \cdot ISO$ and the potential deviation from mean conditions by fluctuations of mixing ratios of both compounds. If the fluctuations are correlated to a certain extent, the quantity $Is=(OH'ISO') \cdot (OH \cdot ISO)^{-1}$ describes the deviation from mean reaction conditions. The possible underestimation of modelled OH or related topics are, therefore, not of relevance to our study. But as mentioned by referee 1, Liu et al. (2018) found "no gap between modelled and observed OH in Amazonia within the experimental uncertainty". Here it is important to note that these researchers did not measure OH directly but inferred OH from other measurements of VOC and NOx.

Therefore the paper by Liu et al. (2018), which is cited by the referee, cannot be used to draw conclusions about the agreement of modeled and measured OH. Liu et al. neither operate a model, nor do they carry out a model-measurement comparison. They show that OH inferred from VOC observations increases with NOy, which would be consistent with predictions by old (MCMv3.2) and new (MCMv3.3) mechanisms as well.

We find $NOx$<0.6 $ppb$ on 22 November 2015 at the ATTO site. For such conditions Liu et al. (2018) inferred $OH \approx 6 \cdot 10^5 \text{ molecules cm}^{-3}$ which is comparable to OH ≈ 0.03 ppt for ATTO 2015 in Fig. S15 – Fig. S16. The total OH reactivity on this day is published by Pfannerstill at al. (2018) and is not only given by isoprene and its oxidation products which, combined, contribute 62 % ± 29 % of the total OH reactivity on this day. Kaser at al. (2013) found a smaller influence of different chemical recycling schemes on mean $OH$ (an average over space and time) but published mean segregation intensity in the range 0.06<$|Is|$<0.15 with maxima up to abut $|Is| \approx 0.3$. Their mean reaction rates $\overline{R_{ij}}$ become reduced by 6% up to 15% (and by about 30% at maximum $I_s$).

Kaser et al. 2015 find that their OH observations tend to be higher than what they calculate by MCM v3.2 and that the implementation of isoprene RO2 recycling in their model and segregation can fill the gap between modelled and observed OH in their campaign. This is also consistent with the most recent isoprene chemical mechanism developments. However, even the most advanced chemical mechanisms cannot explain the high OH observations, which were reported, for example, in Lelieveld et al. (2008) or Hofzumahaus et al. (2009). I think, we need more research and this should include studying the role of segregation between biogenic VOCs and OH.

"Old" mechanisms (e.g. MCMv3.2 and earlier) predict significantly smaller OH concentrations in forest environments compared to "new" mechanisms, which contain additional OH recycling by isomerization of isoprene RO2, such as the MCMv3.3, the Caltech mechanism (Wennberg et al., 2018), or the modified isoprene mechanism proposed by Novelli et al. (ACP, 2020). As Novelli et al. point out, the implementation of the isomerization chemistry can lead locally to an OH enhancement of a factor of 3 at 20 pptv NO. However, if OH is also removed by other VOCs, or when recycling by reaction with NO becomes more relevant, the impact of the isoprene RO2 isomerization is getting smaller. It is essential to note that Kanaya et al. studied the OH measurements in ambient air outside of the chamber, which was the second part of the HOxComp campaign.

For additional information we give in a Supplement in Fig. S15 the OH mixing ratios for ATTO 2015 and ECHO 2003 as function of NOx and further relations in Fig. S16 – Fig. S17.

The way how OH measurements were performed is described in Section 2.2 and will be discussed in more detail in the revised version in this section 2.2 and in a Supplement. Here we will also refer to the background signal and the topic of an "offset problem".

As mentioned above, despite the nice review article of Wennberg et al., 2018, there are still new findings emerging regarding the OH + isoprene chemistry. Novelli et al. (2020) concluded from their chamber measurements in Jülich: "It was found that the MCMv3.3.1 for isoprene degradation initiated by OH radicals is not able to reproduce the measured trace gas concentrations in the experiments despite the inclusion of the isomerization reaction for isoprene-RO2 following the LIM1 mechanism for NO mixing ratios <0.2 ppbv. Large discrepancies are observed, in particular for OH radicals, with a ratio of modelled to

measured OH of 0.7±0.07and of almost a factor of 2 for the sum of MVK, MACR, and ISOPOOHs (all isomers)."

Anyhow, since 2010 we operate the ground based as well as the aircraft based instrument using a 'IPI' system for the chemwave technique (Hens et al. 2014 , Novelli et al. 2014, Mao et al. 2012) to remove ambient OH and to quantify the chemical OH background signal. We did compare our measurements using the IPI with the CIMS in Hyytiälä (Hens et al.) as well as with the CIMS on the Hohenpeissenberg (Novelli et al. 2014) and concluded that the usage of IPI does get our instrument into agreement with the CIMS instruments, which also use a OH scavenger to quantify the chemical OH background signal.

In the introduction we describe some results from literature on OH measurements and their comparison to modelling. We will refer to mean mixing ratios of OH (and $NO_x$) in section 1 and section 2 and in a Supplement (see Figs. S11 – S17) and also in the discussion and comparison of results from the field to results from modelling including the suggested literature.

One possibility for the discrepancy between measurements and modelled mean mixing ratios is the influence of segregation that we directly experimentally asses in this paper. Even if measured and modelled OH is said to be equal in any of the papers mentioned, their difference is in the range of ±20% to ±40% (and often a systematic deviation is given). Note that $I_s$ influences the mean reaction rate; to what extent a mean mixing ratio is influenced can only be stated if the complete balances are considered.

But our studies on segregation were performed basically to find out a) if this phenomenon occurs in the atmosphere and b) which atmospheric processes may be related to $I_s$ especially for OH + ISO.

Note that Liu et al. (2018) did not perform measurements of OH as discussed above. The results given by Kaser et al (2015) are compared to mean box calculations and allow $I_s$ in a range up to about 15% ( and even 30%) (see also above).

Here we discuss deviations from purely homogeneously mixed cases for a chemically reactive compound (isoprene) with OH, as derived from instrumental determination at that time and that specific location, and also compare to results from modelling. We only take published information. The deviation is given by the normalized covariance in Eq.1. To further illustrate our comparisons, we will compare $I_s$ as function of height from different studies (experiments / models) in a Supplement and give a comparison with respect to the $NO_x$ environmental mixing ratios (see our Figs S15 – S17 and Fig. S22).

In addition, we will give errors for the measured quantities in revised Figures in our paper (Fig 5, Fig.6, Fig.7), so that the reader can get information for ECHO 2003 and ATTO 2015. "Commenting on conclusions of the papers by Lin et al. and Kaser et al." will be given in section 1 in a way that we mention what is given in their publications and what is needed to draw a certain conclusion.

*Rev: The derivation of some of the simplifications is poorly explained – eq. 19: why would only one triple term be important in the analysis here? RES, RE and REis are not well explained – I assume REis refers to term I in eq. 21. In general, I miss a thorough analysis of error propagation in context of the presented equations (e.g. eq 21). Many terms are dropped because they are supposedly small, yet the impact of the experimental limitations is not rationalized well in context of the variance budget of isoprene. I suspect that a significant amount of variance of isoprene might not be accounted for due to spectral attenuation. It also appears that the data availability is rather thin – I only count about 16 individual data points for the analysis presented in Fig3. Within the uncertainty of the analysis, I wonder whether this is enough to draw some of the presented conclusions after considering a thorough analysis of the propagation of errors (ie. systematic and random).*

We will refine our explanations in the revised version. These are the data existing so far. To our knowledge, there are not more data points available.

Ans: Eq. 19 has a sign error and, therefore also Fig. 5 – 7 have to be corrected. The correct Eq. 19 reads:

$$I_s + CH_{is} + nvar(ISO)_{is} + RE_{is} = 0$$

This revision of figures is given in the Appendix: „Revised Figures". In addition term $RE_{is}$ needs to be corrected in the revised version.

Errors (from „Error Propagation") for $I_s$ are given for ECHO 2003 and ATTO 2015 results in Figs. 5-7. Following the procedure which was described in Dlugi at al. (2010/ 2014).

Each data Point is for a 10 minute interval and describes one data point of $I_s$ for this interval.

References:

Hens, K., Novelli, A., Martinez, M., Auld, J., Axinte, R., Bohn, B., Fischer, H., Keronen, P., Kubistin, D., Nölscher, A. C., Oswald, R., Paasonen, P., Petäjä, T., Regelin, E., Sander, R., Sinha, V., Sipilä, M., Taraborrelli, D., Tatum Ernest, C., Williams, J., Lelieveld, J., and Harder, H.: Observation and modelling of HOx radicals in a boreal forest, Atmos. Chem. Phys., 14, 8723–8747, https://doi.org/10.5194/acp-14-8723-2014, 2014

Novelli, A., Hens, K., Tatum Ernest, C., Kubistin, D., Regelin, E., Elste, T., Plass-Dülmer, C., Martinez, M., Lelieveld, J., and Harder, H.: Characterisation of an inlet pre-injector laser-induced fluorescence instrument for the measurement of atmospheric hydroxyl radicals, Atmos. Meas. Tech., 7, 3413–3430, https://doi.org/10.5194/amt-7-3413-2014, 2014.

Pfannerstill, E. Y., Nölscher, A. C., Yáñez-Serrano, A. M., Bourtsoukidis, E., Keßel, S., Janssen, R. H. H., Tsokankunku, A., Wolff, S., Sörgel, M., Sá, M. O., Araújo, A., Walter, D., Lavrič, J., Dias-Júnior, C. Q., Kesselmeier, J. and Williams, J.: Total OH Reactivity Changes Over the Amazon Rainforest During an El Niño Event, Front. For. Glob. Chang., doi:10.3389/ffgc.2018.00012, 2018.

---

## Author Comment (AC2) · 20 May 2020

We thank the reviewer for his very detailed and thorough review. We are confident that we addressed all points raised in the below comments. Regarding the major concerns: We carefully checked the manuscript regarding the "inconsistency with negative signs "and revised the manuscript accordingly. The submitted manuscript was focused on the dynamical causes/limitations of Is but we followed the reviewer's suggestion to additionally look into the causes/limitations of Is resulting from the chemical regime which is driven by $NO_x$ abundance. Furthermore, we give some reasoning why we think that the air-craft and modeling study, whose inclusion was criticized, are comparable to our results and are worth including. We used data from the LES model from the near surface layer 10 m to 30 m for the detailed analysis and the CBL-integrated value only for comparison to other CBL-integrated LES studies like in fig. 13. The study of Kaser at al. (2015) also covers the frequency range (i.e. scales) given by the ground based measurements although it extends to larger scales as well. More detailed comments in the following:

In general: We propose to add a Supplement to the revised version of the paper. Here we give figures S1 – S27 for this Supplement as an example to provide information on the micrometeorological and chemical situation at the ATTO site on day 22/11/2015.

Regarding the $NO_x$- dependency:
We compared the dependency of $Is$ on NO, $NO_2$, and $NO_x$ mixing ratios and found no significant relation for the measured data (see Fig. S22 in the Supplement). In addition, we give Fig. S15 – S17 for relations between mean OH and mean $NO_x$, mean isoprene and mean OH and mean Isoprene and mean $NO_x$ (here we also compare to results from Kim et al. (2016)). The results given in Fig. S15 could be compared to those inferred by Liu at al. (2018) as well.

Comparability to Kaser et al. (2015) and Ouwersloot et al. (2011):
According to the Ergodic theorem, one can compare averages over space with averages over time. The link of spatial and temporal scales has been shown for the atmosphere by Orlanski (1975). The scales in space are therefore not the only relevant scales, and only the ranges of frequencies f or wave numbers $k=2\pi f u$ (u= Wind velocity or flight velocity) are important. Kaser et al. (2015) give horizontal length scales from about 3 km up to 220 km. With the mean measuring velocity of the aircraft of 100 ms$^{-1}$ one obtains their frequency range of measured data (OH) between $4.5 \cdot 10^{-4}\ Hz$ and $0.033\ Hz$. Note that we obtained data for $1.7 \cdot 10^{-3}\ Hz$ (respectively $5.4 \cdot 10^{-4}\ Hz$) up to $0.2\ Hz$ (ECHO) respectively $0.067\ Hz$ (ATTO). Kaser et al. (2015) extrapolated their spectra for OH by a direct proportionality to the spectra of $O_3$ (not to the chemical production term of OH, which would include the fluctuations of photolysis frequency and water vapor as well). By this procedure, they extended their spectral presentation to higher frequencies. On average they give $Is \sim 0.13$ with a contribution from the smaller scales ($>5 \cdot 10^{-3}\ Hz$) of 0.06 to 0.08 (in absolute values) which is within the range of our measured surface values. Therefore, the data given by Kaser et al. (2015) directly fit into our Fig. 12. The results from Ouwersloot et al. (2011) are taken from their original LES data from mean vertical profiles for the lowest layers (20 m vertical resolution) to compare to our results from heights above rough surfaces.

*"Much of this work is a direct reworking of the results presented in Dlugi et al. (2014)":*

In our figures Fig. 1 – 14 we present the results of measurements and analysis – and some intermediate steps. We describe what is given by the analysis of data following the concept that terms of balance equations are determined. We don't mix experimental data with modelled data in the analysis of our data in contrast to other sources given in

literature. We then compare the derived results to those form modelling studies. We are aware that not all quantities that are needed to describe this dynamical – chemical – biological system in a complete way (also in the sense of theoretical physics / mathematics) were measured. Therefore, some of them are estimated from additional studies. As this needs to be done for the ATTO data anyway and it is an important step forward to have it done for two contrasting environments ("high-NO$_x$ low isoprene" and "low-NO$_x$ high isoprene") as direct measurements are still very sparse we think that all steps should be included. The point "…to have it all done in one place…" has also been acknowledged by the reviewer as well.

We think it is important not only to focus on quantities that show a "convincing relationship" as a) correlation does not mean causality and b) to give a framework on how data are organized and if different studies can be compared it may be good to see in which state of the system (convection/no convection or dominance of transport/dominance of chemistry) they were derived. Variables describing the state of the system may also be important to decide whether data from different states can be used to derive general relationships between Is and other quantities. This refers especially to Figs. 13 and 14 (Is versus Damköhler number and buoyant production respectively). We do not quite understand the criticism on the relationship between Is and r. The correlation coefficient has been used in modelling studies to derive Is (in absence of other available data) measured. Though derived from very different environments, the ATTO and ECHO data show the same behavior and none of the measured data is below the line with a slope of 2.5. This means that if all data show this behavior even at a perfect anticorrelation (correlation coefficient -1) Is would reach at maximum a value of -0.4. This is an important information from data analysis only. Furthermore, from Fig. 9 it is clear that models tend to have a higher r$_{ij}$ due to the more Gaussian distribution of the data than obtained from field measurements.

Generally, we compared any empirical relations between $Is$ and other quantities. As example we take the concept of shear and buoyant generation (or "production") as illustrated in Fig. 4.24 from R. Stull (2000). Corresponding Richardson numbers are given in our Fig. S5.

We noticed for ECHO that a certain empirical relation exists between the "buoyant production BP" and $Is$ for a range of BP above $3 \cdot 10^{-3}$ $m^2 s^{-3}$, which is related to free convection conditions. This is given in Fig. 14 for ECHO 2003. The results for ATTO 2015 only partly follow this relation (Fig. 14b in revised figures). In contrast to the results for ECHO 2013 the higher BP ($> 3 \cdot 10^{-3}$ $m^2 s^{-3}$) is related to $|-Is| <$ 0.04 because the correlation coefficients become small (see also revised Fig. 9). All these $Is$- data from ECHO 2003 are related to ogives of the covariance $\overline{OH'ISO'}$ with partly negative and partly positive contributions, which sum up to small values of $Is$ only in the time interval of 10 minutes. This is described by Dlugi et al. (2014), e.g., in their Fig. 17.

For ATTO 2015 the $I_s$ values for $BP > 5 \cdot 10^{-3} m^2 s^{-3}$ are below $|I_s| = 0.04$. The contribution to $I_s$ from frequencies higher $5 \cdot 10^{-3} Hz$ is small. Predominantly, eddies from the low frequency range contribute to the covariance $\overline{c_i' c_j'}$ respectively the correlation coefficient $r_{ij}$. Their contribution to $I_s$ is not sufficiently covered by a ten – minute averaging interval during situations where cloud and surface induced vertical and horizontal convective mixing interact. This is also illustrated by Figures S26 and S27 for $I_s$ as function of surface sensible heat flux $H_s$. (Note that $I_s$ is always given at the end of the 10 – minute interval and, therefore, is shifted to the right compared to maxima or maxima in $H_s$.) Therefore, above about $BP = 3.5 \cdot 10^{-3}$ $m^2 s^{-3}$ only one data point shows a larger $I_s$ ($I_s = -0,067$), while the others are influenced by the effect mentioned above. Note that we choose a 10- minute

interval to approach conditions for stationarity (see our remarks on data analysis and covariance calculations). If we would extend the range to lower frequencies, as done for ECHO, stationarity conditions would not be fulfilled.

The relation given in Fig. 13 is again an empirical presentation of the data on $Is$, which shows that for moderately unstable conditions all data point towards an increasing (with increasing Da) but limited $Is$. Some points, where "$u_*$- scaling" is not fulfilled, don't follow such behavior.

Line-by-line concerns:

| | |
|---|---|
| 40 - 43 | Rev: Lines 40-43: Please give an approximate range of NO$_x$ for these conditions (instead of "high NO$_x$" and "low NO$_x$" which probably mean different things to different readers.) As stated above, I believe that it may be the most important distinction among the various experimental results.

 Ans: ECHO 2003: $0.2 \leq NO \leq 1.3 \ (most \ data \ for: \ > 0.75) ppb; \ 1.1 \leq NO_2 \leq 6.5 \ (most \ data \ for: \ > 2.5) ppb$
 ATTO 2015: $NO < 0.5 \ ppb; \ NO_2 < 0.5 \ ppb$ See also $I_s$ as function of $NO$ $(NO_2, NO_x)$ in Fig. S22. |
| 49 - 49 | Rev: l. 48-49: I disagree, a direct relation is shown in your equations (11) and (21). The isoprene flux is contained in two leading terms: one is in the TPI of the covariance budget and the other is in the variance budget (GPvar). But, as you say here, they may be more or less influential depending on the strength of other competing terms.

 Ans: A direct relation is not given because $I_s$ is not directly proportional to the isoprene surface flux, as other terms of the corresponding equation significantly contribute to the result. (see section 4.1)
 In addition: The term in Eq. (9) respectively Eq. (11) (which may be related to the influence of the flux of isoprene on $Is$ in Eq. (3) or Eq. (6)) is the product of the flux and the vertical derivative of the mixing ratio profile (Stull, 1988, p. 133). For an upward directed flux at the surface, in general the mixing ratio decreases with height, and, $\partial c_i / \partial z$ becomes negative. (Convention: An upward directed flux is a loss at the surface, and, therefore, has a negative sign). The (positive) influence of both product terms enter into the Eq. (11) for the variance of isoprene and even if all other terms would vanish, the influence of $\partial c_i / \partial z$ would remain.
 Therefore, the correlation between $I_s$ and the flux decreases as given in Figures 10 and 11 already near the surface and even more in the ABL.
 Considering the balance of the variance, what is often called "a correlation with the flux of isoprene" is the correlation with the variance (or standard deviation) of isoprene. |
| 51 | Rev: l. 51: I realize that this may seem like quibbling, but chemical scalar fluxes do not necessarily always decrease with height. For example, if the entrainment is strong and the CBL concentration high enough, then it is possible for the isoprene vertical flux to increase with height (e.g. water vapor fluxes in some cases.) |

| | |
|---|---|
| | Ans: For isoprene the observations given by Su et al. (2016) for the time and area where Kaser et al. (2015) performed their measurements suggest decreasing fluxes with increasing height. For ECHO 2003 we have fluxes obtained near top of canopy and flux (profiles) decreasing with height (not published).

Here we consider these data sets from measurements only. If entrainment of isoprene – as it is reported for water vapor – would be observed a secondary circulation must have transported isoprene above the inversion and chemical removal must be small, so that entrainment fluxes becomes significant (and advection would be the controlling term). |
| 58 | Ref: l. 58: You should probably be more quantitative about this statement. How much does Is increase with measurement bandwidth? It seems that if this is one of the leading findings of the study, worthy of inclusion in the abstract, then it should be elaborated a bit more: what does the cospectrum of  look like at low wavenumbers? The increase in lower frequencies that you investigate in this study goes from about 2-10 km (based on a wind speed of about 4 m/s), comparable to the LES domain of Ouwersloot et al. (2011). On the other hand, the scales covered in Kaser et al. (2015) start at about 3 km (30 s OH measurements) and run to 50-100 km, which are dramatically larger scales than even your expanded analysis. This is the main reason that I believe the inclusion of the results of Kaser et al. (2015) is not appropriate for this work.

Ans: We refer here (in the short abstract) to the data analysis given in Figs. 12 and 13, but possibly not explicitly enough explained in the text (line 1379 - 1386). We therefore, will add an additional description in a Supplement.
In addition:
In general the contribution from large frequencies to any quantity increases with height. The spectral behavior of ogives (and the interaction of upward and downward transport) was discussed by Dlugi et al. (2014) for $cov(OH, ISO)$ in their Fig. 17. We performed the same analysis as for ECHO also for ATTO and will give examples on these results in a Supplement related to current Fig. 12 and Fig. 13 in the text.
In the Abstract we will clarify the procedure; e.g.: "The spectral contribution to the covariance $cov(OH, ISO)$ and $I_s$ was analyzed. Both quantities increase with increasing contribution of lower frequencies."
Note that the frequency range by Kaser et al. (2015) is about $4.5 \cdot 10^{-4}\,Hz \leq \nu \leq 0.033\,Hz$. It was artificially extended to higher frequencies by postulating a one to one proportionality between OH and $O_3$. We had $1.7 \cdot 10^{-3}\,Hz \leq \nu \leq 0.2\,Hz$ (Dlugi et al. 2010 and 2014) with an extension to $5.6 \cdot 10^{-4}\,Hz \leq \nu \leq 0.2\,Hz$ from the analysis of measured data as given in Table 2 / Fig. 12 -13 and the text below Fig. 12. Therefore, the difference is given mainly in the high frequency range, if no extension would be applied. This shows that a comparison of the data sets is not limited by significant differences in spectral contributions to $cov(OH, ISO)$ although the spatial scales covered are different. See also above comment. |
| 70 | Rev: l. 70: "and" is probably more often the case (buoyant & shear production combined).
Ans: We will replace "and" by "as well as" |
| 160 | Rev: l. 160: missing 'g' in Tg.
Ans: We correct this typing error ("$T_g$") |
| 165 | Rev: l. 165: I don't see why the transport has to take place in a cloud- |

| | |
|---|---|
| | topped boundary layer. |
| | Ans: The experiments ECHO 2003 and ATTO 2015 were performed in a cloud topped ABL. In general, the notion "ABL" also includes clouds. Therefore we write: … in the ABL …. |
| 172 | Rev: l. 172: I don't think that $NO_2$ generally determines the OH reactivity in any significant way. Also, whenever there is any significant isoprene, it tends to be the dominant VOC sink for OH. Therefore it doesn't make sense to consider isoprene moving through a static OH field because isoprene is usually the dominant sink and determines, in part, the OH field. Of course, this rapid reactivity is what drives the anticorrelation. |
| | Ans: Here we give relevant reactants for the chemical interacting cycles not only directly related to OH reactivity. You write that isoprene determines, in part, the OH field. "In part", yes: Pfannerstill et al. (2018) published that the isoprene-related OH reactivity on the same day we made our measurements was 53 ± 29 % (average ± standard deviation) of the total OH reactivity. Therefore, the anti -correlation is not (-1) but significantly below (-0.5). OH is locally produced and destroyed with a chemical lifetime below 1 s, while isoprene has a lifetime of about 300 s or larger. Or in terms of Damköhler number OH has $Da > 20$ for a fast reacting compound while isoprene has $0.01 < Da < 0.5$. We do not consider a static OH field. But the locally variable OH is not transported on a scale above about several centimeters and the variability of OH is given by all chemical sources and sinks on the scales of this small volume. Therefore, we give this conceptual frame as suggested also in Dlugi et al. (2014). |
| 178 | Rev: l. 178: Doesn't Is < 0 hold only for when the chemical sink of OH is dominated by isoprene or something correlated with isoprene? If isoprene is correlated with a major source of OH (e.g. RO2 or HCHO) then Is could be > 0 in principle, no? Come to think of it, this is where I believe the answer to the origin of your observed |Is| limitation lies: the OH photochemistry is so heavily buffered that while isoprene is a dominant sink it is also correlated with important sources such as isoprene peroxy radicals, Iso-O2, and HCHO. See Kaser et al., 2015 Fig. S9 for estimates of the relative magnitudes of ROx (=RO2 + HO2) source strengths. |
| | Ans: With our measurements we try to find out "what can be seen in the atmosphere" or literally spoken "what does nature tell us?". We thought about and discussed the segregation problem in general and in very detail for more than 20 years, but in the introduction of this paper we give the status of results being published. We consciously avoid in this context to mix information with qualitative world views. By definition $0 \leq |I_s| \leq 1$. But why $|-I_s| < 0.3$ and not $|-I_s| > 0.5$? Kaser et al. (2015) give box model results (well mixed conditions) on chemical sources and sinks for OH and suggest which pathways are important, as several (cited) researchers did before. They compared these "box results" to their measurements in a turbulent and convective atmosphere. They do not quantify the causes for turbulent fluctuations and the occurrence of $|I_s| \gg 0$ but state that their data show high $|-I_s|$ over areas with high computed surface fluxes. Here we refer to section 3.3 of Ouwersloot et al. (2011) where they showed that regions of higher isoprene emission may dynamically decouple from the surrounding (their case LSB2) resulting on high $I_s = -0.405$ but with other chemistry above the different spatial parts. Flying over heterogeneous areas may result in detection from strong updrafts |

| | |
|---|---|
| | and weaker downdrafts resulting in a high $I_s \approx -0.3$ by averaging over different parts of the flight track (see Ouwersloot et al. (2011), page 10697). We did not discuss this topic because such conditions cannot be simply related to ECHO 2003/ ATTO 2015. But we will add some more results obtained by Ouwersloot et al. (2011) to our introduction and also to our graphical presentation if we present $I_s$ as function of height. |
| 188 | Rev: l. 188: 'caused by' is an odd way to put it because Is < 0 is by definition represents an anticorrelation, but what actually caused it is the broader question that this paper tries to address.

Ans: Yes, you are right. We changed the text accordingly. |
| 207 | Rev: l. 207: The Is values of Kim et al. (2016), albeit very small, nearly double across the range of $NO_x$ from the experiments you compile (~0.1 from ATTO to ~2 ppb from ECHO) from about -0.02 to -0.035.

Ans: Regarding Fig. 2 of Kim et al. (2016) Is decreases with increasing $NO_x$ (se also our fig. S22) and increases again for the very high $NO_x$ case, where $NO_x$ acts as an OH-sink.
We included $NO_x$ in our analysis (see Figs. S17 and S22 in the Supplement). From the measurements (ATTO and ECHO) there is no observable trend of Is with changing $NO_x$. We will add the intercomparison of near surface measurements related to ECHO 2003 and ATTO 2015 and also refer to Ouwersloot et al. (2011) and their findings in the revised manuscript. |
| 209 | Rev: l. 209: The UNO run of Ouwersloot et al. (2011) developed an Is of -0.12 and it was 'homogenous' in heat and isoprene fluxes, whereas without $NO_2$ (lower $NO_x$) the control run Is = -0.07. Do you mean Is (low $NO_x$) > Is (high $NO_x$) or their magnitudes? Note that Ouwersloot et al. (2011) (from their Section 3.6) "stress the need to take the VOC and $NO_x$ conditions into account in future studies that aim at segregation parameterizations." This advice seems to have been overlooked in the present work.

Ans: The values and cases in this section have been given for comparison and to give some overview of the existing literature. Given the above discussion and the inclusion of the $NO_x$ dependence in our revised manuscript we need to elaborate more on this point. Kim et al. (2016) found a slight decrease of Is in the surface layer by increasing the $NO_x$ surface flux two times by one order of magnitude ("very low $NO_x$" to "High $NO_x$"). This decrease is attributed to higher mean OH that causes lower mean isoprene and therefore lower isoprene fluctuations. Another increase by a factor of 5 brings them to the "very high $NO_x$" case where $NO_x$ becomes a significant OH sink and Is increases again.
Ouwersloot et al (2011) state that a change of one order of magnitude in surface $NO_x$ fluxes did not significantly change Is. The cases discussed here are with 0.5 ppb $NO_x$ in the free troposphere, which is entrained into the BL and causes larger values of Is. These scenarios are different from the conditions given by Kim et al. (2016) and by our surface measurements. We include the discussion of the $NO_x$ dependency into the revised manuscript. A first result is given in Fig. S22.
The UNO run had $NO_2 = 0.5\,ppb$ in the free troposphere and otherwise is comparable to run HOM. $I_s(HOM) = -0.07$, $I_s(UNO) = -0.124$.
Adding spatial heterogeneity in isoprene source strength and moisture and heat fluxes (HNO) leads to $I_s = -0.209$. Without any spatial variability of fluxes of moisture and heat and no $NO_2$ entrained from the free troposphere, but a spatial heterogeneity in the isoprene source |

| | |
|---|---|
| | strength (LSB1) $I_s = -0.07$. Changing the scale of heterogeneity leads to $I_s(LSB2) = -0.405$, $I_s(LSB3) = -0.308$ and $I_s(LSB4) = -0.177$ for example. Here we refer to their discussion on the influence of enhanced $NO_x$ in these model calculations in their section 3.6. Such conditions are completely different from our field studies ECHO 2003 / ATTO 2015 where $NO_x$ sources are located mainly near or at the surfaces. In a Supplement (see Fig. S22) we add $|-I_s|$ versus $NO_x$ for ECHO and ATTO to present the results and to compare with literature and especially results from Kim et al. (2016).) |
| 213 | Rev: l. 213: Again, I disagree with the statement that most of the results of Ouwersloot et al. (2011) are < -0.1. From their Table 4, HOM  = -0.07 which is definitely not < -0.1! (see less important weakness (1) above about comparisons of Is magnitude or numerical values less than zero.)

Ans: Most results for homogeneous cases (and no gradients) are smaller or about $I_s = -0.1$ (<: "smaller than"). For clarification we may write instead $|-I_s| < 0.1$. Any "disturbance" to these cases cause $|-I_s| \geq 0.1$. |
| 243 | Rev: l. 243: I think you should define this here: <w'c'2>. At this point I did not have any idea what M21 represented. Incidentally, this appears to be the best predictor you have observed to correlate with Is, so why not emphasize that more and show the results for M21 vs. Is in the ECHO & ATTO data sets?

Ans: We will give the explicit notation of $M_{21}$ in the text and also add $M_{21}$ versus $I_s$ graphical results for both studies near canopy top in another section of this paper. The corresponding figures are given in the Supplement (Fig. S20, Fig. S21). |
| 255 -257 | Rev: l. 255-257: I think you should point out here that most theoretical treatments show Is to be of smaller magnitudes in the bulk of the CBL vs. the surface layer (e.g., Kaser et al., 2015; Ouwersloot et al., 2011; Patton et al., 2001.)

Ans: In this section of the manuscript we only refer to Kaser et al. (2015): a) Patton et al. (2001) present results only near canopy top also as function of Da: $Da = 0.17$, $I_s = -0.05$; $Da = 0.6$, $I_s = -0.17$ with a very simplified chemistry.
b) Kaser et al. (2015) give $I_s$ as function of height throughout the ABL from model calculations (model used by Patton et al. (2001)) and found significantly larger values of $I_s$ near canopy top, smaller ($0.08 \leq |-I_s| \leq 0.2$) near $z/z_i = 0.5$ in the ABL and larger also near $z_i$.
c) Ouwersloot et al. (2011) give examples on $I_s(z)$ which show such results also.
In order to clarify this point we will add some comments on vertical profiles of $I_s(z)$ from model calculations in line 261. |
| 316 | Rev: l. 316: The measurement of HOx fluxes and other higher moments from July 25 of the ECHO campaign seems like new material (not covered in Dlugi et al., 2014) so it probably merits some more explication. For instance, how did the cospectra compare to <w'T'> or <w'O3'> or some other scalars? Does the flux direction and magnitude agree to theoretical predictions (e.g. Gao & Wesely, 1994)? Were these data analyzed in a separate paper? What changed to allow for these measurements, of which I know of no others? See Section 5.1 of Dlugi et al. (2014): "Spatial derivations of mixing ratios of these compounds and their fluxes are not available from this data set". What has changed? |

| | Ans: The fluxes of radicals and related compounds as presented in Dlugi et al. (2010) for information were not discussed here as this was not the scope of this manuscript which is already quite extensive. "What has changed"? Some of us in 2017 (M. Berger, M. Zelger, R. Dlugi, G. Kramm) (re-) analyzed additional calibrated data from ECHO 2003 to allow at least better estimates of terms in the budget equations, which were not evaluated in 2013 / 2014. We applied some of these results in this paper. Most results on reactive compounds and their fluxes from ECHO 2003 are not published, although they were evaluated. (see also line 740 - 746). |
|---|---|
| 321 -323 | Rev: l. 321-323: What seems more directly important than temperature and humidity is the mean concentrations of HOx on that day relative to the rest of the experiment.

Ans: Temperature and humidity conditions also influence chemistry. But we will add OH and $NO_x$- mixing ratios in the text and in the Supplement (see Fig. S10 – Fig. S13 and Fig. S15 – Fig. S16). |
| 345 | Rev: l. 345: If the HOx data is available at 0.2 Hz, why were there no fluxes reported other than July 25?

Ans: As described in Dlugi et al. (2010), at the end of their section 1, "a one-day feasibility study was performed" and segregation could be calculated from these measurements for OH + isoprene, OH + monoterpenes the first time for atmospheric conditions and fluxes for $HO_2$ are given as well. |
| 378 | Rev: l. 378: What was an average background OH value relative to the total? And as far as the second unit is concerned, how are you sure the amount of background OH is the same in both units? It seems that it might be worthwhile to describe some statistics of the backgrounds for both units to understand their variability and similarity.

Figure 1: This figure is probably not necessary since none of the analysis of the OH measurements is covered herein.

Ans: We give more details in the revised version. Fig. 1 will be completed to show distances between measuring volume of the eddy system and the different inlets which are needed for calculation of time shifts for determination of covariances and mixed higher moments. |
| 453 | Rev: l. 453: Equation 4 is a few pages later than this reference, but come to think of it equation 4 does not really give any chemical information other than there is a reaction between OH and isoprene, and as such is probably not necessary.

Ans: Eq. (4) gives what is done in this paper, and, only therefore is given a number. |
| 457 | Rev: l. 457: The justification of time resolution invoked by Karl et al. (2013) is for turbulent fluxes (from Lenschow & Kristensen, 1985), which relies on assumptions about the turbulent statistics of w', but the requirements for a covariance with another scalar are different.

Ans: This reference is given by Kaser et. al. (2015) in their paper and we cite only what is written in their paper and in their supplement. A discussion of their results in other sections is also only based on their reported findings. |

| 489 -491 | Rev: l. 489-491: I don't think it makes sense to mention compressible fluids and refer to the various 'a' variables defined by Richardson only to redefine them.

Ans: We did it because to name the definitions clarifies the theoretical framework on which most atmospheric measurements are based. This is not trivial, but often forgotten. Segregation therefore is given in the "Reynolds" description of fluids, applied also in most atmospheric models. The averaging procedure and choice made to replace averages by time averages is given to refer to these aspects of "point" measurements which "see" some of the spatial variability of their surroundings. Higgens et al. (2013) only recently discussed this "Ergodic hypothesis" and compared it to experimental findings. |
|---|---|
| 500 | Rev: l. 500: I do not think that ''' is the best symbol to use for multiplication? It looks like a cross product and the subscripting i,j,k looks like tensor notation. I think a '×' or nothing at all is much more conventional and clear to indicate scalar multiplication.

Ans: "x" is given by ACP for multiplication. The further notation follows the theoretic framework given – for example – by Vilá – Guerau de Arellano and Vinuesa or Verver in their modelling papers. ($i \equiv ISO$, $j \equiv OH$ and k, l, m, n are index for physical variables) |
| 507 | Rev: l. 507: Very rapidly after the H abstraction of the OH + isoprene reaction the production of some isomer of an Iso-O2 radical occurs. These peroxy radicals reacting with NO are usually a very important source of OH (Kaser et al., 2015 estimate it to be of similar magnitude as ozone photolysis.)

Ans: We analyze the measured data in terms of Eq. (3) and further equations. Of this chemical cycle, the first step is analyzed and evaluated. If some compound is not measured, we cannot analyze it within this concept. We know that peroxy radicals exist, but only $HO_2$ was directly measured (and analysis of segregation - e.g. of $HO_2 + NO -$ has started) |
| 508 | Rev: l. 508: Is this supposed to mean Eq. 2 subtracted from Eq. 1?
This was misleading it should be "and". |
| 515 | Rev: l. 515: What exactly is CxHyOz? I recognize the attempt to remain general, but I think the more important general species that is not mentioned anywhere is RO2.

Ans: We give the general form, which includes RO2. : $RO_2 \cong C_xH_yO_2$ with $R = C_xH_y$ |
| 529 | Rev: l. 529: the denominator of term 1 should be rwci.

Ans: Yes in Eq. (6). |
| 574 | Rev: l. 574: Using the Einstein summation convention with tensor notation confuses your own convention of using a generalized chemical trace gas, ci and cj. These look like 3D vectors in tensor notation (i, j = 1, 2, 3), and there is no other place in the manuscript where it is beneficial to generalize the chemistry. This is a work centered on the reaction of OH and isoprene and as such there is nothing gained by calling those species cj and ci, respectively. For example, kij, looks like a second order tensor, not a scalar reaction rate coefficient.

Table 1: It would be more clear if you preserved the signs of these terms |

| | |
|---|---|
| | such that MR & TR are always <0 (that is, act to reduce <ci> in the budget.) Realizing that you labeled the terms inside the parentheses in equation (7), the terms DMF and DTF have different units because they are not the divergence thus the numbers in the table do not have to sum. I think it would be a lot clearer if you defined the terms to each entire item in the budget equation and keep their signs clear and indicative of how they change <ci>.

Ans: Chemical compounds have index i, j, other (e.g., wind velocity vector components) have k, l, m following – for example - Vinuesa or Vilá – Guerau de Arellano. $k_{ij}$ is explained in the text as all other quantities.
Table 1: We will write – for example – (for S) - $-0.8 \cdot 10^{-3} \ to \ 1.2 \cdot 10^{-3}$, and all other quantities for clarity.
In Eq. (7) we will write each term separated from each other, so that formally dimensions are the same. DMF and DTF are the spatial derivatives and dimensions are identical for all terms as for MR and TR and S. |
| 603 -628 | Rev: l. 603-628: This entire paragraph suspiciously omits any mention of horizontal components of these advective terms. That, in itself, seems like an oversight, and it renders the last sentence (l.626-628) incorrect: a change in mean horizontal advection (without a change in the wind field divergence) can lead to significant changes in <ci>. It is not clear whether you mean 3D or 2D divergence/convergence in this discussion. Keep in mind that the second term in equation (8) in its full 3 dimensions is zero because of the incompressibility of the mean flow.

Ans: We add the discussion of horizontal advection influences. |
| 707 | Rev: l. 707: Doesn't the concentration of isoprene decrease with height directly above the canopy making the numbers you report -0.01 to -0.07 ppb m-1? This term then is always negative acting to reduce the steady-state variance, no? This is counterintuitive, but is the nature of your steady-state approximation in equation (11).

Ans: Yes, we introduce the sign for clarification. The term $GP_{var}$ is given by the product of the flux and the corresponding "gradient". The source of heat, water vapour or isoprene is "at the surface". The surface (the plants at the surface) sees a loss as the flux is directed upwards. Therefore the convention is to consider this flux to have a negative sign (see – for example – the results given by R. Stull (1988 respectively 1993) in "An Introduction to Boundary Layer Meteorology" for variances of $\theta_v$ (page 113) or moisture (page 130). [For momentum the same concept holds.] |
| 717 | Rev: l. 717: I think you should be more specific in this reference. I believe it is specifically discussed in Section S3.2 of the Dlugi et al. 2014 supplementary materials.

Ans: We will expand our discussion on this topic because Dlugi et al. (2010) described events with downdrafts and Dlugi et al. (2014) in more detail horizontal advection. We will give both situations with explanation of the specific conditions (and its possible influence on differences of mixing ratios $\Delta C$) and the same estimates for ATTO 2015. |
| 723 | Rev: l. 733: Fig. 1 of Spirig et al. (2005) indicates a tower separation of ~250 m.

Ans: Spirig et al. (2005) performed measurements at the west tower and |

| | |
|---|---|
| | the main tower (225 m away from the west tower). A third tower was installed in a distance of 120 m to the East from the main tower. So the correct number is 225 m – as determined by A. Schaub (co- author) to support the physical modelling by Aubrun (2005) instead of 125 m (typing error) in our text. The calculations were done with 225 m. |
| 740 -746 | Rev: l. 740-746: TTvar is the divergence of a turbulent flux of variance. Speaking of a "vertical change of TTvar" sounds like you are now looking at the second derivative of the variance flux. Is that what you're referring to? It would help if this discussion were a lot more clear about what is a turbulent flux of variance (the <w'c'2> term), vs. its vertical change (TTz,var).

Ans: This was indeed misleading and we refer here to the difference in the variance flux with height. We got a proportionality between both terms for isoprene and $\Theta$, and, on two days also results for two heights above canopy for ISO, $\Theta$, and q. These results are applied to estimate $TT_{var}$ for isoprene. We will give a more detailed explanation in the revised version (Supplement).
We clarified the text. (see also answer to comment on line 316). |
| 752 | Rev: l. 752: Your term III in Eq (11) is equivalent to IV3 in Table 4 of Dlugi et al. (2014) which states its estimated magnitude as < 3e-5 ppb2 s-1.

Ans: We did a reanalysis of all these data for ECHO 2003 and corresponding calculations for ATTO 2015. But indeed the given exponent is wrong for both experiments: $< 10^{-5} ppb^2 s^{-1}$ is the correct information as not all values are below $10^{-6} ppb^2 s^{-1}$. |
| 762 | Rev: l. 762: It is not clear how you estimate the gradient of a fluctuating scalar directly, but in general variance budget discussions the molecular destruction term is expected to be first order (to balance mean gradient production in the steady-state, flow-integrated condition.) See Section 5.3 of Wyngaard (2010), for example.

Ans: The $c_i$ (fluctuations) were measured on the given days for several hours in two heights above canopy with time resolution of 1 Hz and synchronization of better than 0.005 s. The equation yields correct dimensions: $10^{-5} m^2 s^{-1} \cdot (ppb^2 m^{-2}) = 10^{-5} ppb^2 s^{-1}$ (see also answer to comment on line 316). |
| 805 | Rev: l. 805: How did you derive these OH flux values? And are you imply that you have these values for both experiments? Is this not discussed anywhere else in the literature? It seems like a very difficult measurement to directly make by eddy covariance. In any event, you should probably specify the sign of this flux (I believe it should be downward, <0). These magnitudes seem much larger than predicted by Gao & Wesely (1994).

Ans: In section 2 we describe that we measured time resolved OH mixing ratios (0.2 Hz for ECHO respectively 0.067 Hz for ATTO) and wind velocity components u, v, w and could filter the w- series to 0.2 Hz respectively 0.067 Hz for calculate $\overline{w'OH'}$ for both field studies.
OH- fluxes are explained in this section and they cannot be applied to calculate a deposition flux, they are given only by term S and MR(TR) in Eq. (7) by the influences of chemical sinks and sources on S and MR(TR).
The effective distance for physical transport is just a few cm, but the |

| | |
|---|---|
| | numerical values are caused by the source and sink distributions of OH. (In line with Gao and Wesly 2004). |
| 834 -835 | Res: l. 834/5: Again, the gradients of isoprene should be negative.

Ans: Yes, we give always the sign in the revised version. |
| 845 | Res: l. 845: I recommend sticking to a single format for all of these range limits of your scale analysis, and preferably using only one significant digit. For example, 'xe-3' to 'ye-1'. Two significant digits for these scale analyses that typically span multiple decades just seems unnecessary and slightly confusing.

Ans: The referee is right and we will give only one digit in the revised manuscript. |
| 859 -861 | Res: l. 859-861: The similarity you are applying to associate the different scalar transport terms needs to be explicitly stated. It seems like you are using some sort of modified Bowen ratio analog to the transport term, but this seems highly speculative. I believe that speculative is fine, but it would be more convincing if you explicitly stated the similarity you are applying.

Ans: Our description needs to be clarified:
We will express especially our own data and give the argument clearer in terms that the heat flux and the turbulent transport of the heat flux are calculated from measurements and are compared to results from measurements of isoprene in two heights above canopy on some days (see above) to calculate comparable terms. A proportionality is obtained for the term III for heat and ISO which is used for this estimate. Term IV is calculated as described in the text. See also answer to comment on line 316). |
| 893 | Rev: l. 893: This range of a factor of 5 for the pressure transport term implies that the time scale values have a range of a factor of 6, because the isoprene fluxes mentioned above span a factor of 30 (0.02 to 0.6 ppb m/s). It would be clearer if you presented what the mixing length concept of Poggi et al. (2004) depended on.

Ans: We will add a short description of this mixing length concept (see Poggi, 2004) in the revised version. |
| 973 | l. 973: I have tried and tried and redone the arithmetic on the governing equation (15), because I know how pernicious and elusive sign errors can be, but I just cannot see how the normalized variance term in equations (20 & 21) can have the opposite sign of the Cij term (which is the balance of the terms from Rij outside of the covariance and variance terms that all have the same sign). This same error appears in Dlugi et al. (2014) at their equation (15). This has very important bearing on the analysis because the normalized variance of isoprene and the RES (Eq. 16) terms both act to increase the magnitude of the OH and isoprene segregation coefficient, in this case, - Is, because Is < 0. It seems like this equation will change the authors' calculations of REis because they solve for it as the residual of equation (21), and will fundamentally change Figure 7.

Ans: We thank the reviewer very much for identifying this sign error by his careful analysis. Starting with Eq. (15) a sign was incorrectly transferred so that Eq. (21) has a wrong sign. The correct form reads $I_s + CH_{is} + nvar(ISO)_{is} + RE_{is} = 0$.
As mentioned in the text for the revised figures the presentations are |

| | |
|---|---|
| | made with $I_s + CH_{is} + nvar(ISO)_{is} = -RE_{is}$.
The revised Fig. 5 – Fig. 7 also correct results presented by Dlugi et al. (2014) with respect to the magnitude of $RE_{is}$ and the relation between $I_s$ and $nvar(ISO)_{is} - (-RE_{is})$. |
| 999 | Rev: l. 999: You say that Rij goes negative despite terms (b) and (c) which are positive definite. But Rij is defined with a negative (definite) sign (equation 18), so the positive definite terms like (b) and (c) work to make Rij negative. I found this language error typical throughout the manuscript. When revising I recommend being very careful with the language about discussing relative values or magnitudes of values, always retaining the accurate signs of the terms.

Figure 3: 60% of graph has no information on it. Also, why is the total term in one unit (ppb2 s-1) and the individual components in another (ppb3). I think it makes the figure less clear to include the reaction rate in one and eliminate it in the others.
Figure 4: Again, why compare these terms of differing units and then put a one-to-one line on the figure? Also why ignore the sign of Rij? If all the values are negative, then label it –Rij

Ans: $R_{ij}$ can be positive or negative and may change sign (Fig. 3) for ATTO 2015. The revised Fig. 3 is given in the Appendix. The text has been clarified according to the comment. |
| 1041 | Rev: l. 1041: Term (c) is not the only leading term of Rij. The ATTO results could differ because of a substantially different contribution from the <OH'Iso'>[Iso] term (a), especially at higher values of [OH]var(Iso).

Ans: The revised Fig. 3 shows that term c is dominant together with term a. (The latter is used to formulate a diagnostic Eq. (21) for $I_s$). As term a) is negative and term c) is positive (for ATTO) their difference determines the presentation in Fig. 3 with partly different results for ATTO and ECHO. |
| 1109 | Rev: l. 1109: "Rij increases [in magnitude] with increasing variance…"

Ans: We will add "in magnitude" |
| 1126 | Rev: l. 1126: It does not seem clear from Fig. 9 that the relationship between rij and Is is non-linear. You use this term a lot but none of the figures clearly show any distinction among a linear or non-linear relationship.

Ans: A numerical fit to the data will be added to show this relation. |
| 1041 | Rev: l. 1041: Term (c) is not the only leading term of Rij. The ATTO results could differ because of a substantially different contribution from the <OH'Iso'>[Iso] term (a), especially at higher values of [OH]var(Iso).

Ans: The revised Fig. 3 shows that term c is dominant together with term a. (The latter is used to formulate a diagnostic Eq. (21) for $I_s$). As term a) is negative and term c) is positive (for ATTO) their difference determines the presentation in Fig. 3 with partly different results for ATTO and ECHO. |
| 1109 | Rev: l. 1109: "Rij increases [in magnitude] with increasing variance…"

Ans: We will add "in magnitude" |
| 1126 | Rev: l. 1126: It does not seem clear from Fig. 9 that the relationship between rij and Is is non-linear. You use this term a lot but none of the |

| | |
|---|---|
| | figures clearly show any distinction among a linear or non-linear relationship.

Ans: A numerical fit to the data will be added to show this relation. |
| 1219 - 1221 | Rev: l. 1219-1221: You do not know for certain that the var(Iso) and flux terms are only established near the surface (for example, the entrainment zone can possess high variances and fluxes.) Furthermore, equation (11) also shows that var(Iso) is augmented by a term proportional to [Isoprene]$<c_i \times c_j>$ (concordant with equation (21) with the corrected sign), which could also be a leading term near the surface. Also note that what you are referring to as GPvar actually serves to decrease isoprene variance in the steady-state form you present in equation (11) because d[Iso]/dz < 0. In the variance budget, equation (9), GPvar produces variance, but in the reactive chemical steady-state of equation (11) it reduces variance.

Ans: We discuss the related terms with respect to the application to our field studies near the surface and some discussion of results from literature (e.g. Kaser et al. 2015).
The upward directed "fluxes" have a negative sign (they describe a sink at the surface; see definition in Stull (1988)). Also $(\partial c_i / \partial z) < 0$ (as the reviewer states), so that the product is positive in both Equations. We refer to the height dependence of different terms as cited in lines 1205 – 1212. |
| 1243 | Res: l. 1243: I am assuming you mean vertical advection by the mean flow. However, just because W is larger in magnitude at higher elevation in the CBL does not mean that the magnitude of the scalar gradient is larger. It is much more likely to be turbulent transport that is a large term. If by 'vertical advection' you mean turbulent transport (the divergence of a vertical turbulent flux), then I would specify that.

Ans: Here we refer to arguments given to your comments line 1331 – 1333 (see below). In addition the term $A_{var}$ in Eq. (9) (respectively Eq. (11)) can be of influence on the magnitude of variance. |
| 1244 | Rev: l. 1244: Is is related to the isoprene flux by two separate terms of Eq. (21): the TPI term of REis and the GPvar term in the normalized variance, nvar(Iso). This is not made clear in this discussion and consequently these arguments are ambiguous. These two flux terms have different coefficients (OH and isoprene gradients, respectively) so that their coefficients will change with altitude (probably both decreasing with height.) I would suggest eliminating all of this height dependence of variance discussion because it is speculative (for reactive scalars) and it does not really help the overall work in any way that I can discern.

Ans: Here we refer to arguments given to your comments line 1331 – 1333 (see below). In addition: we applied the convention that upward directed fluxes are a sink of isoprene (or heat, or moisture) at the surface and a gain for the atmosphere. Therefore, upward directed fluxes have a (-) sign. Thus, the term $GP_{var}$ becomes positive in Eq. (9) as well as in Eq. (11). |
| 1319 | Rev: l. 1319: No, OH and O3 do not necessarily have a large positive covariance (presumably someone could check if there were O3 fluxes being measured on the tower), but the principal source of OH (on the ~1 s time scale) is the photo dissociation of O3 so it is very likely that they are, in fact, correlated. |

| | |
|---|---|
| | Ans: Yes, we agree; here we only repeat the arguments originally given by Kaser et al. (2015). To clarify we quote the text in "_". |
| 1331 - 1333 | Rev: l. 1331-1333: That is patently incorrect. First, Ei0 is directly related to the flux at any height in the CBL (you used such a relationship yourself earlier to extrapolate their observed fluxes at z/zi ~ 0.4 to the surface). Furthermore, as stated previously, Is is correlated to the isoprene flux through both the GPvar (where it serves to diminish the variance, and thus \|Is\|, right above the canopy where the flux is upward and the gradient is negative), and in the TPI term of REis in (21) where it tends to be a source of negative covariance because the OH gradient is likely positive (due to preponderance of sinks effusing out of the canopy.)

Ans: As described in the text the relation termed "no longer valid" is that Is can be approximated by the isoprene (surface) flux, not that the isoprene flux at a certain level is not related to the surface flux (this was just one condition to be meet to establish this relationship). The whole paragraph describes that one can establish such relationships, but that the predictive power of the surface values diminishes with increasing height in the ABL as other terms become more important in the budgets.
We applied the CBL scaling as described in lines 1195 – 1203. This concept considers the ABL a slab with boundaries at bottom and at top and the emission flux at bottom and the entrainment flux at top. An emission flux (like the sensible heat flux) decreases with height – represented like a flux divergence; the sensible heat flux crosses the zero line, if entrainment is observed (see some summary in Sorbjan (1989)).
Here the change of mixing ratio with time is proportional to the flux divergence as given in the balance of the mixing ratios (Eq. (7)). Thus the flux decreases with height. In line 1205 – 1221 we focus on term $GP_{var}$ from the balance of variance, because of Eq. (3) – the definition of $I_s$ . We argue that the contribution of the term, which is the product of the isoprene flux and the (vertical) gradient, decreases with height, and therefore the correlation coefficient obtained at the surface between the isoprene flux and $I_s$ decreases with increasing height. Therefore, findings given by Kaser et al. (2015) and in our paper are consistent.
The term $TPI$ in $RE_{is}$ in Dlugi et al. (2014) is the product of the isoprene flux and the spatial gradient of the OH mixing ratio. At least the isoprene flux decreases with height. Above canopy top we observed a decrease of OH with height for ECHO 2003. For ATTO 2015 we have no data on the vertical OH profile. At least for ECHO this influence on $I_s$ decreases with height. Due to your remarks we will relate the discussion of $GP_{var}$ and $TPI$ in the revised version.

Rev:
Figure 12: Why are there are not the same number of blue diamonds (spectrally extended) as there are black circles? They should be 18-27% larger according to line 1391. Also, the blue diamonds all lie exactly on top of the circles showing no spectral change in <OH'Iso'>. Also, the two points on the lower left (Is < -0.2) do not seem to exist on Fig. 14.

Ans: The calculation of $I_s$ for 30 minutes intervals instead of 10 minutes intervals is related to the calculation of the covariance but also the calculation of the product of the means. Both quantities changed compared to the results for 10 minute intervals resulting in less values of |

| | |
|---|---|
| | the covariance.
If results for 30 minutes overlap with a value obtained for 10 minutes this does not mean that the related values (belonging to the 30 minutes interval, 3x10 = 30) are not larger by the given percentage increase. The overlapping was caused by the choice of resolution of the graphic software, which was set too low (see revised fig. 12). We corrected this so that "overlapping" is avoided and explain this in the text below that figure. The blue dashed curve gives the empirical range for all data. |
| 1409 – 1410 | Rev: l. 1409-1410: Is never becomes independent of [Iso][OH] because that product resides in its denominator. The covariance may become independent, but not Is.

Ans: For covariances smaller than -4x10$^{-5}$ (larger negative numerical value) $mean(OH) \cdot mean(ISO)$ stays approximately constant. Therefore the increase of $I_s$ becomes only controlled by the covariance. We clarified the text accordingly. |
| 1460 | Rev: l. 1460: According to Dlugi et al. (2014) Eq. (17) M12 were considered the "ejections", and M21 the "sweeps"?

Ans: This was a typing error. Furthermore, we add the results of Is versus M_21 as Fig. S20 – Fig S21 to the Supplement.

Rev: Figure 14: This figure is nearly identically the same as Dlugi et al. (2014) Figure 20, save for the three modeling results and two Kaser et al. (2015) points. Why do you not present any of the ATTO data on this figure? Why plot both BP and kinematic heat flux? As far as I can discern there is no appreciable difference in the underlying relationship and plotting both just clutters the figure.

Ans: We add the ATTO data to this presentation (see Fig. 14b - bottom). The value $BP > 3 \cdot 10^{-3} \, m^2 s^{-3}$ is the range of the onset of free convective conditions. We added this discussion to the text. This is a physical criterion, so no statistical comparison is made because we sort data (of $I_s$) in terms of the surface sensible heat flux and the buoyant production term. |
| 1545 | Rev: l. 1545: It is very challenging to find an empirical relationship in Fig. 14 as stated. You should propose one if you think it exists. Is looks to me like a nearly vertical line would fit through the points of BP > 3e-3? I wonder what the p-value of such a fit would be, because it does not look great by my eye.

Ans: There is no direct correlation of Is versus heat flux expected, but as thermals are known to have a profound influence on Is it is important to know if data points were derived in a convective regime or not. This graph is not meant to find a predictor, but to sort the data (see also general comments).
If we consider the concept of the shear and buoyant generation (or production) we noticed that a certain empirical relation exists between the buoyant production BP and $I_s$ (and the sensible heat flux and $I_s$). For small BP, $I_s$ is not dependent on BP, but if a value of $3 \cdot 10^{-3} \, m^2 s^{-3}$ is approached $I_s$ – values increase. Here the data are in the range where the free convection limit is approached, which is about $3 \cdot 10^{-3} \, m^2 s^{-3}$ following for example results given by Stull (2000). Model results from Ouwersloot at al. (2011) and $I_s$ from ECHO follow this behavior. The |

| | |
|---|---|
| | results from ATTO 2015 increase if this "limit" is approached, but become smaller than $\|-I_s\| = 0.04$ for $BP > 3 \cdot 10^{-3}\ m^2 s^{-3}$. In this range the correlation coefficient in Eq. (3) becomes small (see also the revised figures Fig. 14a, Fig. 14b and Fig. 9). |
| 1597 -1601 | Rev: l. 1597-1601: If M21 vs. nvar(Iso) & REis shows a strong relationship as in Fig. 18 of Dlugi et al. (2014) why not show it? If this finding is worthy of a paragraph in conclusion, then it seems it should be represented in a figure. Earlier you state the sweeps only weakly correlate with nvar(Iso) and REis, and here you state that only ejections contribute to Is. This all seems to beg for a figure of both M21 and M12 to see how much they each correlate to nvar(Iso) & REis. This could be a micrometeorological parameter that is readily measured in canopy field studies that could be used to estimate Is for chemical modelers, for example.

 Ans: We give $I_s$ versus $M_{21}$ in the Supplement figures (Fig. S20 and Fig. S21). |
| 1621 | Rev: l. 1621: The bandwidth of the Kaser et al. (2015) measurements were out to nearly 100 km. For typical winds speeds of, say, 5 m/s this would require a 5.5 hr integration time at a tower site. Thus the measurements, aside from being made several hundred meters higher than the ECHO & ATTO datasets, represent a much larger spectral band. The 'hypothesis' of scale dependence is established explicitly in Ouwersloot et al. (2011), why bring this in as a hypothesis from this work? There is currently no easy way to disentangle the isoprene surface source variability from the scale of the measurements in terms of their effects on Is, so it is not a hypothesis that is truly tested in this work.

 Ans (see also general comments): Kaser et al. (2015) give horizontal length scales from about 3 km up to 220 km. With the mean measuring velocity of the aircraft of 100 m s$^{-1}$ one obtains their frequency range of measured data (OH) between $4.5 \cdot 10^{-4}\ Hz$ and $0.033\ Hz$. Note that we obtained data for $1.7 \cdot 10^{-3}\ Hz$ (respectively $5.4 \cdot 10^{-4}\ Hz$) and $0.2\ Hz$ (ECHO) respectively $0.067\ Hz$ (ATTO). Kaser et al. (2015) extrapolated their spectra for OH by a direct proportionality to the spectra of $O3$ (not to the chemical production term of OH, which would include the fluctuations of photolysis frequency and water vapor). By this procedure they extended their spectral presentation to higher frequencies. On average they give $Is \sim 0.13$ with a contribution of the range $>5 \cdot 10^{-3}\ Hz$ by absolute values of -6% to -8%.
 The scales in space are not of interest, only the ranges of frequencies f or wave numbers $k = 2\pi f u\prime$ (u= Wind velocity or flight velocity) are important. Therefore the data given by Kaser et al. (2015) directly fit into our Fig. 12. The results from Ouwersloot et al. (2011) are taken from their original LES data from mean vertical profiles for the lowest layers to compare to our results from heights above rough surfaces. The Ergodic theorem tells us that one can compare averages over space with averages over time. |
| 1625 -1627 | Rev: l. 1625-1627: This is an interesting idea, but not very well explicated in the body text of the manuscript, and only sprung on the reader in the last sentence of the work. The diurnal source correlations (which in and of itself would promote a positive species covariance) occur on long time scales relative to the chemistry and the TKE dissipation and the 10-40 minutes averaging used in this study. In order for this to be a reason for the "limits" of Is suggested on the 10 min scale the sources would need to correlate on this restricted time scale, and/or |

there would need to be some sort of downscale cascade at play. This speaks to the absence of any cospectral representation of Is in this work (something like Fig. S4 of Kaser et al., 2015), which would help understand its spectral dependence. In any event, I suspect the compensating chemistry of OH sources that are correlated with isoprene (e.g. isoprene peroxy radicals) are the most likely culprits for limiting the magnitude of Is.

Ans: This part of the "summary and conclusions" will be revised.
 The "restricted time scale" is only one aspect, but if extended, the absolute value of $I_s$ increases. If we consider the results obtained by Ouwersloot et al. (2011), the strategy how to sample over up- and downdrafts becomes important. We analyze "ogives" to present these results in the text (with information also in the supplement), to give limits to $I_s$ from the mixing processes. In this context we will also discuss possible limits to $I_s$ occurring from chemical cycling.